# Chemical characteristics of cloud water and the impacts on aerosol properties at a subtropical mountain site in Hong Kong

Tao Li[1,2], Zhe Wang[3], Yaru Wang[2], Chen Wu[1,2], Yiheng Liang[2], Men Xia[2], Chuan Yu[2], Hui Yun[2], Weihao Wang[2], Yan Wang[1], Jia Guo[4], Hartmut Herrmann[1,5] and Tao Wang[2]

[1] School of Environmental Science and Engineering, Shandong University, Qingdao 266237, China
[2] Department of Civil and Environmental Engineering, The Hong Kong Polytechnic University, Hong Kong, China
[3] Division of Environment and Sustainability, The Hong Kong University of Science and Technology, Hong Kong, China
[4] Research Center for Eco-Environmental Sciences, Chinese Academy of Sciences, Beijing 100085, China
[5] Atmospheric Chemistry Department (ACD), Leibniz Institute for Tropospheric Research (TROPOS), Permoserstrasse 15, 04318 Leipzig, Germany

*Correspondence to*: Zhe Wang (z.wang@ust.hk)

**Abstract.** To investigate the cloud water chemistry and the effects of cloud processing on aerosol properties, comprehensive field observations of cloud water, aerosols, and gas-phase species were conducted at a mountaintop site in Hong Kong in October and November 2016. The chemical composition of cloud water including water-soluble ions, dissolved organic matter (DOM), carbonyl compounds (refer to aldehydes and acetone), carboxylic acids, and trace metals was quantified. The measured cloud water was very acidic with a mean pH of 3.63, as the ammonium (174 µeq L$^{-1}$) was insufficient for neutralizing the dominant sulfate (231 µeq L$^{-1}$) and nitrate (160 µeq L$^{-1}$). Substantial DOM (9.3 mgC L$^{-1}$) was found in cloud water, with carbonyl compounds and carboxylic acids accounting for 18% and 6% in carbon molar concentrations, respectively. Different from previous observations, concentrations of methylglyoxal (19.1 µM) and glyoxal (6.72 µM) were higher than that of formaldehyde (1.59 µM). The partitioning of carbonyls between cloud water and the gas phase was also investigated. The measured aqueous fractions of dicarbonyls were comparable to the theoretical estimations, while significant aqueous-phase supersaturation was found for less soluble monocarbonyls. Both organics and sulfate were significantly produced in cloud water, and the aqueous formation of organics was more enhanced by photochemistry and under less-acidic conditions. Moreover, elevated sulfate and organics were measured in the cloud-processed aerosols, and were expected to contribute largely to the increase in droplet-mode aerosol mass fraction. This study demonstrates the significant role of clouds in altering the chemical compositions and physical properties of aerosols via scavenging and aqueous chemical processing, providing valuable information about gas–cloud–aerosol interactions in subtropical and coastal regions.

## 1 Introduction

Clouds in the troposphere play a key role in atmospheric aqueous-phase chemistry by acting as efficient media for the in-cloud formation of sulfate and secondary organic aerosol (SOA) (Harris et al., 2013; Ervens, 2015). Numerous studies on cloud and fog chemistry have been conducted in Europe and North America since the 1990s (Collett et al., 2002; Ervens, 2015;

van Pinxteren et al., 2016). During the past decade, studies of the compositions of cloud/fog water, cloud scavenging and aqueous-phase reactions have also been carried out in Asia, particularly in China and Japan (Aikawa et al., 2007; Guo et al., 2012). In-cloud sulfate production, which causes acid rain, has been extensively characterized (Lelieveld and Heintzenberg, 1992; Harris et al., 2013; Guo et al., 2012). Recently, more attention has been given to organic materials, which are present in

comparable amounts as sulfate and nitrate in cloud and fog water (Collett et al., 2008; Herckes et al., 2013), because of the significant contribution of chemical cloud processing to aqueous SOA (aqSOA) formation in high-humidity environments (Ervens et al., 2011; Huang et al., 2011; Tomaz et al., 2018).

Many field observations and laboratory studies have reported direct evidence for the in-cloud formation of low-volatile products and aqSOA. Kaul et al. (2011) observed enhanced SOA production and increased ratios of organic to elemental

carbon (OC/EC) upon fog evaporation due to aqueous-phase chemistry. Comparison of the mass spectra of ambient aerosols and cloud organics suggests that functionalization of dissolved organics possibly dominates the formation of SOA through oxidative cloud processing (Lee et al., 2012). A chamber study showed faster SOA formation (by a factor of 2) from isoprene photo-oxidation under cloud conditions than dry conditions (Brégonzio-Rozier et al., 2016), highlighting the importance of aqueous-phase reactions. Aircraft measurements by Sorooshian et al. (2007) found ubiquitous layers of enhanced organic acids

levels above clouds, implying that the in-cloud formation of organic acids contributes significantly to emerging organic aerosol layers after droplet evaporation. Oxalate, an aqueous-phase oxidation product, has been considered as a good tracer for aqSOA formation, given the common in-cloud formation pathway of oxalate and sulfate (Yu et al., 2005; Sorooshian et al., 2006). Single-particle mass spectrometry analysis confirmed that oxalate in cloud droplet residuals and cloud interstitial particles was three times as abundant as that in ambient (cloud-free) particles (Zhang et al., 2017), demonstrating the in-cloud formation of

oxalate. At present, soluble dicarbonyls are recognized as the primary precursors of carboxylic acids and oligomers in the aqueous phase (Lim et al., 2010; Ervens et al., 2011). The irreversible uptake and aqueous oxidation of glyoxal, the simplest dicarbonyl compound, is suggested to be the primary formation pathway of oxalic acid and aqSOA (Warneck, 2003; Carlton et al., 2007). In general, water-soluble organic compounds (e.g., carbonyls) can partition into cloud droplets and form low-volatility products such as carboxylic acids and oligomers, which stay in the particle phase after cloud evaporation and form

aqSOA (Blando and Turpin, 2000; Lim et al., 2005; van Pinxteren et al., 2005; Carlton et al., 2007; Lim et al., 2010; Galloway et al., 2014; Brégonzio-Rozier et al., 2016).

Chemical cloud processing not only contributes to aerosol mass production but also alters the chemical composition of aerosols. Highly oxidized aqSOA usually exhibits higher O/C ratios (1–2) compared to SOA formed in the gas phase (0.3–0.5) (Ervens et al., 2011), as indicated by model predictions that glyoxal SOA formed in cloud water and wet aerosols are

predominantly oxalic acid and oligomers, respectively (Lim et al., 2010). Even with similar O/C ratios, the molecular compositions of organics in aerosols and cloud water could be quite different, for example, the organosulfate hydrolysis and nitrogen-containing compounds formation were observed in cloud water compared to atmospheric particles, suggesting the significant role of cloud processing in changing the chemical properties of aerosols (Boone et al., 2015). In addition to the in-cloud sulfate formation (Meng and Seinfeld, 1994), the in-cloud organics formation is also likely to add substantial mass to

droplet-mode particles (Ervens et al., 2011). For example, maximum droplet-mode organics and a shift in particle mass size distribution were observed in simulated cloud events (Brégonzio-Rozier et al., 2016). However, our current knowledge of aqSOA formation mechanisms and how aerosol properties change during real cloud processing remains limited.

The Hong Kong and Pearl River Delta (PRD) region is one of the most industrialized areas in Asia, and experiences serious

particulate and photochemical air pollution. High cloudiness and abundant water vapor lead to significant gas–cloud–aerosol interactions in this region, and half of all surface SOA are estimated to be contributed by the aqueous chemistry of dicarbonyls (Li et al., 2013). To better understand the role of chemical cloud processing in aerosol production and the associated changes in its physicochemical properties, we conducted a comprehensive field campaign with simultaneous measurements of trace gases, aerosols and cloud water at a mountaintop site in Hong Kong. In this paper, we first present an analysis of the chemical

composition of cloud water and then discuss the partitioning of individual carbonyl compounds between gaseous and aqueous phases. Finally, cloud water organics formation and the effects of cloud processing on aerosol physicochemical properties are investigated.

## 2 Methodology

### 2.1 Observation site and sampling

The field campaign was carried out at the summit of Mt. Tai Mo Shan (Mt. TMS, 22°24'N, 114°16'E, 957 m a.s.l.), the highest point of Hong Kong in the southeastern PRD region (Wang et al., 2016), where the coastal and subtropical climate leads to frequent occurrence of cloud/fog events. The site is influenced by both urban/regional pollutions from the PRD region and cleaner marine air masses from the western Pacific Ocean. Cloud water, aerosols, and gas-phase carbonyl compounds were simultaneously sampled from 9 October to 22 November 2016.

Cloud water samples were collected in a 500 mL acid-cleaned HPDE bottle by using a single-stage Caltech Active Strand Cloudwater Collector (CASCC) with a flow rate of 24.5 $m^3$ $min^{-1}$. A detailed description of the collector can be found in our previous work (Guo et al., 2012). The sampling duration was set to 1–3 hours to obtain enough sample volume. Cloudwater pH and electrical conductivity were measured on site using a portable pH meter (model 6350M, JENCO). After filtration through 0.45 μm microfilters (ANPEL Laboratory Technologies (Shanghai) Inc.), aliquots of the cloud water samples for

dissolved organic carbon (DOC, 30 ml), carbonyl compounds (20 ml), water-soluble ions (15 ml), organic acids (15 ml, add 5% (v/v) chloroform added) and trace metals (15 ml, 1% (v/v) hydrochloric acid added) were properly prepared and stored at 4 °C in the dark until laboratory analysis. Derivatization of carbonyl compounds (refer to aldehydes and acetone) with 2,4-dinitrophenylhydrazine (DNPH) was performed on site after adjusting the pH to 3.0 using a buffer solution of citric acid and sodium citrate. Interstitial gas-phase carbonyl compounds were sampled with acidified DNPH-coated silica cartridges (Waters

Sep-Pak DNPH-silica) at a flow rate of 0.5 L $min^{-1}$ for 2–4 hours using a semi-continuous cartridge sampler (ATEC Model 8000). A Teflon filter assembly and an ozone scrubber were installed before the cartridge to remove large droplets and particles and prevent the influence of ozone. All cartridges were refrigerated at -20 °C after sampling.

Daily fine aerosol samples were collected on quartz filters (47 mm diameter, Pall Inc.) using a four-channel sampler (Thermo Anderson, RAAS-400, USA) with a size-selective inlet removing particles/droplets larger than 2.5 μm, with a flow rate of 16.7 L min$^{-1}$ and sampling duration of 23 hours. The sample filters were then refrigerated at -20 °C before laboratory analysis. An ambient ion monitor (URG 9000) with a 2.5 μm cut-size cyclone inlet was used to measure the hourly concentrations of water-soluble ions in $PM_{2.5}$. During the cloud event, the collected fine aerosols were most of interstitial aerosols, because almost all activated cloud droplets were larger than 3 μm and were removed by the cyclone in the sampling inlet. A NanoScan SMPS nanoparticle sizer (Model 3910, TSI Inc.) and an Optical Particle Sizer (OPS) spectrometer (Model 3330, TSI Inc.) were used to measure particle mass size distributions in the range of 0.01 to 9.05 μm with 29 size bins at 1-minute scan intervals. Because of the instrument test and failure, the valid data for ambient ions and particle size distribution were only available from 2 to 11 November and 3 to 21 November, respectively. Trace gases including $SO_2$, $NO_x$, and $O_3$ were measured with a pulsed UV fluorescence analyzer (Thermo, Model 43c), a chemiluminescence analyzer (Thermo, Model 42i) and a UV photometric analyzer (Thermo, Model 49i), respectively. Hourly $PM_{2.5}$ mass concentration data were provided by the Hong Kong Environmental Protection Department. Ambient temperature and relative humidity were measured using a MetPak Weather Station (Gill, UK), and the solar radiation was monitored using a spectral radiometer (Meteorologie Consult GmbH, Germany).

## 2.2 Laboratory chemical analysis

Water-soluble organic carbon (WSOC) in $PM_{2.5}$ sample filters was extracted with 20 ml Milli-Q water (18.25 MΩ cm, Millipore) via sonication for 30 min and then filtration. The DOC in cloud water and WSOC in $PM_{2.5}$ were quantified by nondispersive infrared detection of $CO_2$ after thermocatalytic oxidation at 650 °C using a TOC analyzer (Shimadzu TOC-L, Japan). Sucrose standards were used for calibration, with a method detection limit of 0.112 mg L$^{-1}$. In this study, the dissolved organic matter (DOM) in cloud water and water-soluble organic matter (WSOM) in fine aerosols were estimated to be 1.8 times of DOC and WSOC, respectively (van Pinxteren et al., 2016).

The DNPH-derivatives of carbonyl compounds in the cloud water samples were extracted into 20 ml dichloromethane for three times. The extract was then concentrated to dry yellow powder by reduced pressure distillation at 38 °C and transferred into a volumetric flask using 2 ml high-pressure liquid chromatography (HPLC) grade acetonitrile. The sampled cartridge of gas-phase carbonyl compounds was similarly eluted with 2 ml HPLC grade acetonitrile to a volumetric flask. The cloud water and cartridge extracts were analyzed using HPLC system (PerkinElmer 200 Series) equipped with a UV detector. The method detection limits were determined to be 0.22 μM for formaldehyde, 0.02 μM for acetaldehyde, 0.13 μM for acetone, 0.13 μM for propanal, 0.05 μM for butanal, 0.09 μM for iso-pentantal, 0.06 μM for p-tolualdehyde, 0.07 μM for glyoxal and 0.15 μM for methylglyoxal. The recovery rate ranged from 81% to 98% for individual carbonyls.

Concentrations of water-soluble ions ($Na^+$, $NH_4^+$, $K^+$, $Mg^{2+}$, $Ca^{2+}$, $Cl^-$, $NO_3^-$ and $SO_4^{2-}$) in the cloud water samples were measured using an ion chromatograph (Dionex, ICS 1000). Four carboxylic acids (acetic, formic, pyruvic and oxalic acids) were analyzed using an ion chromatograph (Dionex, ICS 2500), with an IonPac AS11-HC separator column under NaOH

gradient elution. Trace metals including Al, V, Cr, Mn, Fe, Ni, Cu, As, Se, Cd, Ba and Pb were measured by inductively coupled plasma mass spectrometry (ICP-MS, Agilent 7500a) based on the EPA 200.8 method. More details on the ions and trace metal analyses were described in our previous works (Guo et al., 2012; Li et al., 2015).

### 2.3 Aqueous phase fraction of carbonyl compounds

The measured fraction of carbonyl compounds partitioning in the aqueous phase ($F_{me}$) is calculated by Eq.1,

$$F_{me}=\frac{C_{cw}}{C_{cw}+C_{int}} \qquad (1)$$

where $C_{cw}$ is the air equivalent concentration of carbonyl compounds in cloud water, μg m$^{-3}$; and $C_{int}$ is the interstitial gas-phase carbonyl compounds concentration, μg m$^{-3}$.

Assuming equilibrium, the theoretical aqueous phase fraction ($F_{theo}$) can be calculated from the following equation (van
Pinxteren et al., 2005),

$$F_{theo}=\frac{K_H \cdot R \cdot T \cdot LWC \cdot 10^{-6}}{1+K_H \cdot R \cdot T \cdot LWC \cdot 10^{-6}} \qquad (2)$$

where $K_H$ is the Henry's law constant, M atm$^{-1}$; $R$ is the gas constant of 0.08205 L atm mol$^{-1}$ K$^{-1}$; $T$ is the mean temperature in K; and LWC is the cloud liquid water content, g m$^{-3}$.

## 3 Results and discussion

### 3.1 Characterization of cloud water chemistry

Thirty-two cloud water samples in six cloud events were collected at Mt. TMS in Hong Kong during the campaign (Figure S1). The average LWC was 0.26 g m$^{-3}$ with a range of 0.08–0.53 g m$^{-3}$. Cloud water pH ranged between 2.96 and 5.94 with a volume-weighted mean (VWM) value of 3.63, lower than the cloud and fog pH observed in most other areas (e.g., Mt. Tai: 3.86 (Guo et al., 2012); Baengnyeong Island: 3.94 (Boris et al., 2016); Lulin mountain, Taiwan: 3.91 (Simon, 2016);
southeastern Pacific: 4.3 (Benedict et al., 2012); and Mt. Schmücke, Germany: 4.30 (van Pinxteren et al., 2016)), indicating the severe acidification of cloud water in this region.

### 3.1.1 Overview of chemical composition of cloud water

Table 1 summarizes the VWM concentrations of water-soluble ions, DOC, carboxylic acids, carbonyl compounds and trace metals in the cloud water samples. The concentrations of sulfate, nitrate and ammonium ions were 231, 160 and 174 μeq L$^{-1}$,
respectively, accounting for 81% of the total measured ions. The sulfate and nitrate concentrations were much lower than those in clouds in northern China (Guo et al., 2012) and in fogs at Baengnyeong Island (Boris et al., 2016), but higher than those at many sites in America, Europe and Taiwan (Straub et al., 2012; van Pinxteren et al., 2016; Simon, 2016). Meanwhile, there was insufficient ammonium to neutralize the acid ions, as indicated by the low slope (0.46) of charge balance between [$NH_4^+$] and [$NO_3^- + SO_4^{2-}$]. The elevated Cl$^-$ (109 μeq L$^{-1}$) and Na$^+$ (69 μeq L$^{-1}$) indicated the considerable influence of maritime air

from the western Pacific Ocean. In contrast to the commonly observed chloride depletion in coastal cloud water (Benedict et al., 2012), the molar ratio of $Cl^-/Na^+$ (1.86) at Mt. TMS was obviously higher than the sea-salt ratio (1.16). The abundant $Cl^-$ in cloud water can be ascribed to potential anthropogenic chloride sources (e.g., coal-fired power plants, biomass burning) in the PRD region (Wang et al., 2016). Non-sea-salt sulfate (nss-$SO_4^{2-}$) was determined to be $96 \pm 3\%$ of total sulfate based on the $SO_4^{2-}/Na^+$ molar ratio in seawater (0.06), demonstrating that $SO_4^{2-}$ was mainly derived from in-cloud oxidation of $SO_2$ (Harris et al., 2013; Guo et al., 2012) rather than marine source.

**Table 1. Concentrations of inorganic and organic species in cloud water samples measured at Mt. TMS during November 2016.**

|  | Unit | VWM | Average | Min | Max |
|---|---|---|---|---|---|
| pH | - | 3.63 | 3.87 | 2.96 | 5.94 |
| $Na^+$ | $\mu eq\ L^{-1}$ | 69 | 93 | 4 | 447 |
| $NH_4^+$ | $\mu eq\ L^{-1}$ | 174 | 235 | 1 | 1413 |
| $K^+$ | $\mu eq\ L^{-1}$ | 4 | 8 | BDL | 54 |
| $Mg^{2+}$ | $\mu eq\ L^{-1}$ | 15 | 23 | BDL | 105 |
| $Ca^{2+}$ | $\mu eq\ L^{-1}$ | 14 | 49 | BDL | 661 |
| $Cl^-$ | $\mu eq\ L^{-1}$ | 109 | 138 | 0.3 | 617 |
| $NO_3^-$ | $\mu eq\ L^{-1}$ | 160 | 238 | 4 | 1285 |
| $SO_4^{2-}$ | $\mu eq\ L^{-1}$ | 231 | 305 | 3 | 1340 |
| DOC | $mgC\ L^{-1}$ | 9.3 | 12.9 | 2.0 | 108.6 |
| Formic | $\mu M$ | 10.8 | 17.1 | 0.2 | 201.8 |
| Acetic | $\mu M$ | 7.2 | 10.2 | 0.6 | 88.2 |
| Pyruvic | $\mu M$ | 1.5 | 2.7 | 0.2 | 22.7 |
| Oxalic | $\mu M$ | 8.3 | 10.3 | 7.6 | 17.5 |
| Formaldehyde | $\mu M$ | 1.59 | 2.10 | BDL | 6.35 |
| Acetaldehyde | $\mu M$ | 0.03 | 0.04 | BDL | 0.11 |
| Acetone | $\mu M$ | 0.76 | 0.77 | BDL | 2.42 |
| Propanal | $\mu M$ | 0.26 | 0.34 | BDL | 1.42 |
| Butanal | $\mu M$ | 0.08 | 0.09 | BDL | 0.19 |
| iso-pentanal | $\mu M$ | 5.90 | 7.05 | 0.63 | 22.9 |
| p-tolualdehyde | $\mu M$ | 0.36 | 0.39 | BDL | 1.16 |
| Glyoxal | $\mu M$ | 6.72 | 9.00 | 0.73 | 47.9 |
| Methylglyoxal | $\mu M$ | 19.1 | 26.7 | BDL | 45.0 |
| Al | $\mu g\ L^{-1}$ | 131.9 | 180.2 | 23.2 | 737.8 |
| V | $\mu g\ L^{-1}$ | 7.9 | 9.5 | 0.2 | 35.7 |
| Cr | $\mu g\ L^{-1}$ | 0.7 | 1.2 | BDL | 5.0 |
| Mn | $\mu g\ L^{-1}$ | 5.9 | 10.9 | 0.9 | 42.6 |
| Fe | $\mu g\ L^{-1}$ | 50.6 | 106.5 | BDL | 316.8 |
| Ni | $\mu g\ L^{-1}$ | 7.1 | 7.7 | 0.2 | 33.0 |
| Cu | $\mu g\ L^{-1}$ | 10.0 | 17.3 | BDL | 85.9 |
| As | $\mu g\ L^{-1}$ | 6.7 | 7.4 | 0.7 | 22.2 |

| | | | | | |
|---|---|---|---|---|---|
| Se | µg L$^{-1}$ | 1.9 | 2.6 | 0.1 | 11.5 |
| Cd | µg L$^{-1}$ | 0.5 | 0.8 | BDL | 2.9 |
| Ba | µg L$^{-1}$ | 3.0 | 7.2 | BDL | 25.1 |
| Pb | µg L$^{-1}$ | 18.7 | 23.2 | 0.2 | 117.9 |

BDL: below detection limit

DOC concentrations varied from 2.0 to 108.6 mgC L$^{-1}$ with a VWM value of 9.3 mgC L$^{-1}$, lower than those in polluted urban fogs but much higher than most remote and marine clouds (Herckes et al., 2013; van Pinxteren et al., 2016; Ervens et al., 2013; Benedict et al., 2012). The VWM concentrations of formic, acetic, pyruvic and oxalic acids were measured to be 10.8, 7.2, 1.5 and 8.3 µM, respectively, accounting for $6 \pm 2\%$ (molar ratio of carbon) of the DOC in total. Carbonyl compounds (Table 1) comprised $18 \pm 10\%$ of DOC in cloud water. Methylglyoxal (19.1 µM) was the predominated carbonyl species, followed by glyoxal (6.72 µM), iso-pentanal (5.90 µM) and glycolaldehyde (3.56 µM), while formaldehyde (1.59 µM) and acetaldehyde (0.03 µM) were much lower. The nearly triple abundance of methylglyoxal compared to glyoxal at Mt. TMS differed from the previous observations at Puy de Dôme, France (Deguillaume et al., 2014), Mt. Schmücke, Germany (van Pinxteren et al., 2005), and Davis, USA (Ervens et al., 2013), where glyoxal concentrations were 2 to 10 times higher than methylglyoxal, but was similar to the results observed at Whistler, Canada (Ervens et al., 2013) where the methylglyoxal/glyoxal ratio was much higher. These different patterns could partially be attributed to the large differences in precursors at various locations (Table S1) and also the availability of oxidants. Generally, the overall yields of these aldehydes from isoprene are much lower than those from the oxidation of aromatics (Ervens et al., 2013) and the latter also contributes to higher yields of methylglyoxal than glyoxal (Ervens et al., 2011). For example, the glyoxal and methylglyoxal yields from toluene are approximately equal (0.14 and 0.12, respectively, at high NO$_x$ conditions), and methylglyoxal yields from xylene exceed the ones of glyoxal by a factor of 6 (0.08 and 0.47, respectively) (Nishino et al., 2010; Ervens et al., 2011). The higher aromatics concentrations (toluene of 2.3 ppb, xylene of 0.9 ppb) than the biogenic isoprene (0.16 ppb) measured at Mt. TMS are expected to be the important precursors of these aldehydes and lead to the different ratio observed in the cloud water. The less cloud water formaldehyde is likely associated with the deficient partitioning of formaldehyde in the aqueous phase as discussed in Section 3.2.

Aluminium (131.9 µg L$^{-1}$) dominated the cloud water trace metals, of which the concentration was comparable to that measured at other mountain sites in China (99.7 to 157.3 µg L$^{-1}$) (Li et al., 2017). Transition metals Fe, Cu and Mn, which play important roles in the heterogeneous catalytical formation of sulfate (Harris et al., 2013), were also found to be abundant in the cloud water, with mean concentrations of 50.6, 10.0 and 5.9 µg L$^{-1}$, respectively. The toxic Pb concentration in cloud water (18.7 µg L$^{-1}$) was around ten times higher than that observed at sites in Europe (1.4 µg L$^{-1}$) (Fomba et al., 2015) and America (0.6 µg L$^{-1}$) (Straub et al., 2012), probably due to traffic emissions from the surrounding city-cluster. Relatively high concentrations of V (7.9 µg L$^{-1}$) and Ni (7.1 µg L$^{-1}$) implied notable impacts of residual oil combustion from shipping emissions (Viana et al., 2009; Wang et al., 2014). Clearly, the cloud water at Mt. TMS was significantly influenced by anthropogenic emissions.

### 3.1.2 Comparisons among different air masses

Three-day back trajectories were reconstructed using the HYbrid Single-Particle Lagrangian Integrated Trajectory (HYSPLIT) model to investigate the origins of air masses arriving at Mt. TMS, which were influenced by both continental and marine air masses. Three types of air mass plumes for the six cloud events (E.1–6) are identified and displayed in Figure 1: continental (E.1–2), mixed (E.3–4) and marine (E.5–6). Detailed descriptions are given in Table S2.

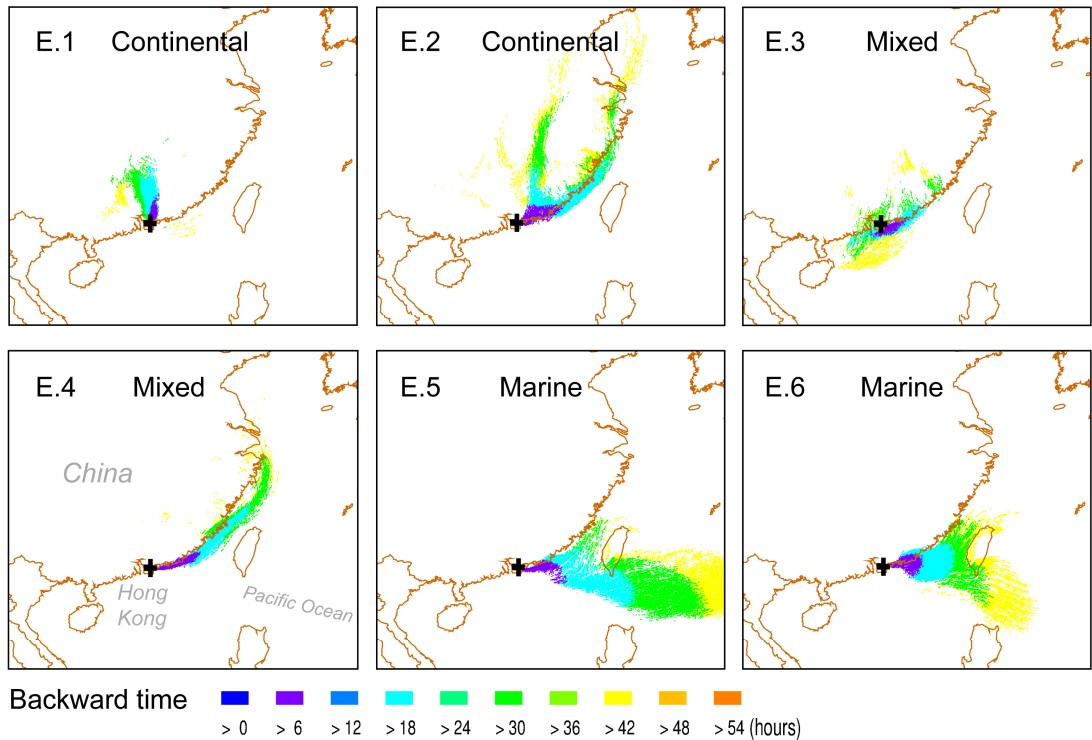

**Figure 1. Air mass plumes arriving at Mt. TMS (black cross) in Hong Kong for six cloud events (E.1–6) simulated using the HYbrid Single-Particle Lagrangian Integrated Trajectory (HYSPLIT) model. The model was driven by archive meteorological data from the Global Data Assimilation System (GDAS, 1° × 1° resolution).**

The concentration and distributions of major components in cloud water during six cloud events are compared in Figure 2a and Figure S2. In general, continental air masses brought more abundant major components including DOM, $SO_4^{2-}$, $NO_3^-$ and $Ca^{2+}$ compared with marine ones, which had lower total concentrations but higher proportions of $Cl^-$ and $Na^+$. For example, continental E.1, which was heavily polluted by anthropogenic emissions within the passage of a cold front (Table S2 and Figure S2), exhibited the largest amount of major components (393.9 mg $L^{-1}$) whereas marine E.6 had the least (15.7 mg $L^{-1}$). For each event, DOM dominated the major components (29–53%), followed by $SO_4^{2-}$ (17–28%) and $NO_3^-$ (17–30%). Nss-$SO_4^{2-}/NO_3^-$ ratios in E.1 (1.03) and E.3 (0.91) were lower than in other events (1.38–1.69), indicating the strong influence of regional air masses from the PRD region. The elevated $NO_x$ from traffic emissions in the HK-PRD region (Zheng et al., 2009) is likely to be responsible for the higher nitrate proportions and lower nss-$SO_4^{2-}/NO_3^-$ ratios in these two events. $Ca^{2+}$ mainly

existed in continental cloud water and the 3% of $Ca^{2+}$ in E.1 likely contributed to the higher pH (5.50). Influenced by marine air masses, the concentration (and proportions) of $Cl^-$ and $Na^+$ notably increased from 0.2 mg $L^{-1}$ (0.4%) and 1.0 mg $L^{-1}$ (2%) in continental cloud water (E.2) to 2.5 mg $L^{-1}$ (5%) and 5.9 mg $L^{-1}$ (11%) in the marine one (E.5), respectively. Meanwhile, the equivalent molar ratios of $Cl^-/Na^+$ and $Ca^{2+}/Na^+$ decreased from 3.11 and 5.06 to 1.50 and 0.04, respectively, close to their ratios in seawater (Table S2). Figure 2a shows elevated proportions of V were observed in marine-influenced E.4–6, which is consistent with plumes passing over the busy international shipping routes (Figure S3), suggesting the contribution of residual oil combustion by shipping to coastal cloud water chemistry (Gao et al., 2016).

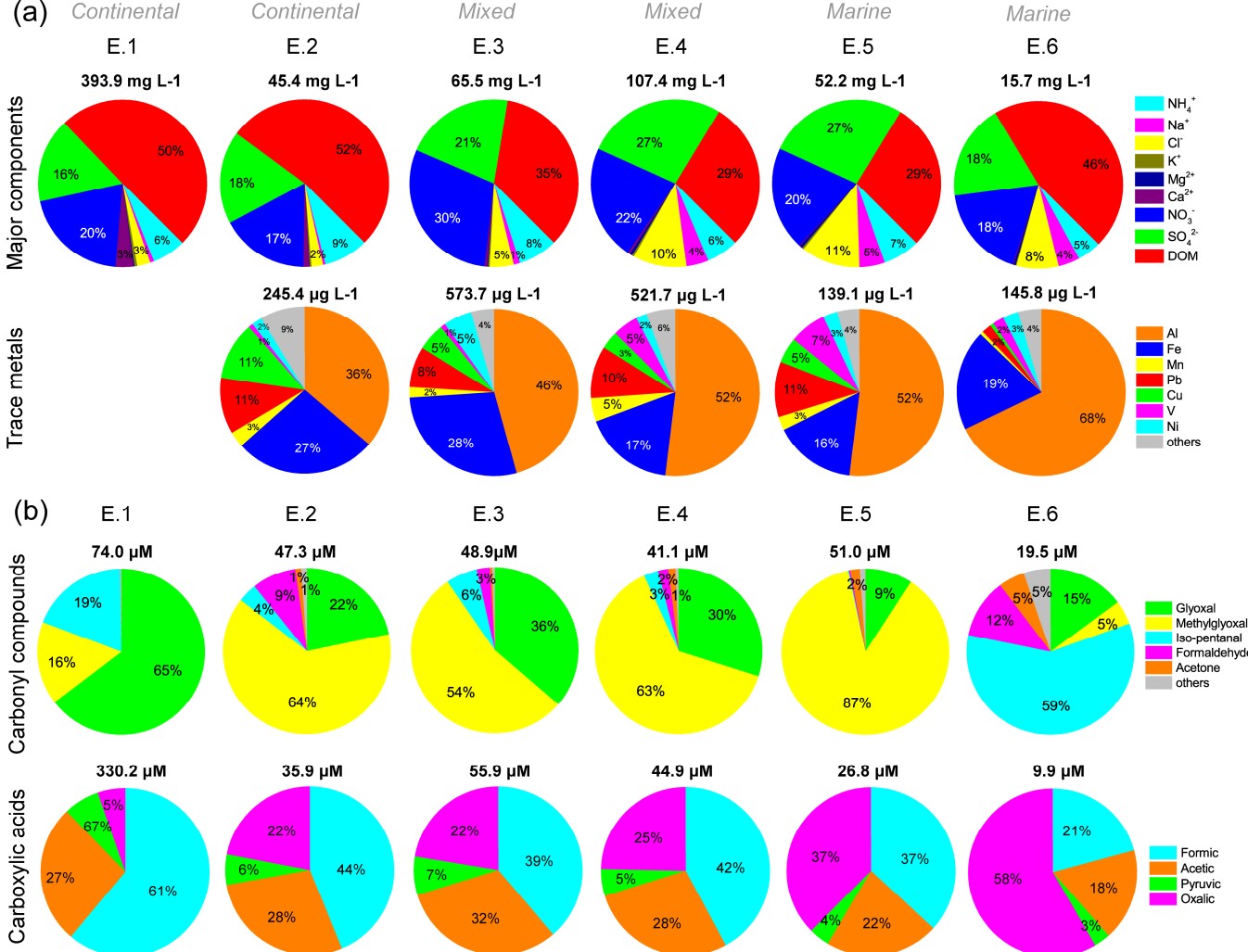

**Figure 2. Concentration distributions of (a) major components and trace metals, and (b) carbonyl compounds and carboxylic acids in cloud water for each cloud event (E.1–6). The volume-weighted mean concentrations of individual species are used. Percentages of carbonyl compounds and carboxylic acids in DOC are determined by carbon molar concentration. Trace metals are absent from E.1 due to limited sample volume.**

Similar trends for the carbonyl compounds and carboxylic acids with major components can also be seen in Figure 2b, but the distribution patterns are obviously distinct. Methylglyoxal dominated the carbonyl compounds in E.2–5, accounting for 54–87% of total carbonyls. In contrast, glyoxal (65%) became the major species in E.1, followed by iso-pentanal (19%) and methylglyoxal (16%); meanwhile, iso-pentanal (59%) was dominant in E.6, which had more glyoxal (15%) than methylglyoxal (5%). The concentration ratios of formaldehyde/acetaldehyde (C1/C2) and acetaldehyde/propanal (C2/C3) in the gas phase during E.3–6 were calculated (Table S3), to diagnose the possible sources of carbonyls in cloud events. The C1/C2 ratios in the range of 2.8–4.5 suggest the combined contributions of both anthropogenic emissions and biogenic sources to the measured carbonyls, because C1/C2 ratios are normally 1 to 2 for urban areas but close to 10 for the rural forests, due to more photochemical production of formaldehyde than acetaldehyde from natural hydrocarbons (Servant et al., 1991; Possanzini et al., 1996; Ho et al., 2002). As propanal is believed to be associated only with anthropogenic emissions, the C2/C3 ratio, which is high in rural atmosphere and low in polluted urban air, can be used as an indicator of anthropogenic origin of carbonyl compounds (Possanzini et al., 1996). The average C2/C3 ratios recorded for Mt. TMS were $4.7 \pm 2.7$, similar to those measured in roadside and urban environments in Hong Kong ($5.0 \pm 0.8$) (Cheng et al., 2014), indicating the considerable anthropogenic sources (e.g., vehicle emissions) of carbonyls at Mt. TMS. The higher concentrations and proportion of iso-pentanal in E.1 (14.02 μM, 19%) and E.6 (11.37 μM, 59%) than in other events were also noted, possibly resulting from unconfirmed direct sources.

The formic-to-acetic acid (F/A) ratio has been suggested to be a useful indicator of sources of carboxylic acids from direct emissions (e.g., anthropogenic sources, biomass burning) or secondary photochemical formation in the gas phase (Talbot et al., 1988), rainwater (Fornaro and Gutz, 2003) and cloud water (Wang et al., 2011b). Direct anthropogenic emission of acetic acid from vehicle-related sources is higher than of formic acid, resulting in F/A ratios much less than 1.0, whereas photochemical oxidation of natural hydrocarbons leads to higher concentrations of formic acid than acetic acid, and therefore the increase in F/A ratios (> 1.0) (Talbot et al., 1988; Fornaro and Gutz, 2003). The F/A ratio in the liquid phase (rainwater or cloud water) is expected to be higher than in the gas phase at equilibrium conditions, which is dictated by Henry's law constants, dissociation constants of formic and acetic acids and pH. So the corresponding gas-phase F/A ratio can be calculated from the aqueous concentrations to evaluate the dominant sources. In this study, a remarkable correlation between formic and acetic acid ($r = 0.97$, $p < 0.01$) suggests their similar sources or formation pathways. The high F/A ratios (1.2–1.9) for E.2–5 (Table S2) indicate the more important secondary formation for carboxylic acids in cloud water. In contrast, the F/A ratios in E.1 and E.6 were 0.4 and 0.5, respectively, suggesting the significant contributions from direct emissions during these two events. In addition, despite the decrease of total concentrations, the proportion of oxalic acid notably increased from 5% to 58% under more influence of marine air masses.

### 3.1.3 Relationships of cloud water composition with LWC and pH

LWC and pH are important factors influencing the phase partitioning, chemical reactions and solute concentrations in cloud water (Tilgner et al., 2005; Li et al., 2017). Figure 3 and Figure S4 show the relationships of individual chemical species with

LWC and pH. The non- and semi-volatile species in cloud water at Mt. TMS including water-soluble ions, DOC, carboxylic acids and trace metals were inversely related to LWC in empirical power functions due to dilution effects, which have been widely observed in previous studies (Herckes et al., 2013; Li et al., 2017). Similar inverse-power relationships of water-soluble ions, DOC and carboxylic acids with pH were also found (Figure 3 and Figure S4). Increased air pollution and secondary acid ions formation likely made the cloud water more acidic, in turn promoting the dissolution of trace metals (Li et al., 2017). Similarly, glyoxal concentrations decreased in a power function as LWC and pH increased. However, methylglyoxal showed different relationships with LWC and pH. Unexpectedly, methylglyoxal tended to increase linearly with LWC and RH for most of the samples except three low-concentration samples (Figure 3d and Figure S4b), which show the opposite pattern to the dilution effect and suggest more methylglyoxal dissolved in diluted and less acidic cloud waters. The different relationships (increase or decrease) of other carbonyls concentrations with increased LWC and pH are also shown in Figure S4. In addition to aqueous-phase reactions, the aqueous/gas phase partitioning of each carbonyl compound influenced by LWC and pH is another possible reason for the observed relationships (Lim et al., 2010; Ervens et al., 2013; Ervens et al., 2011).

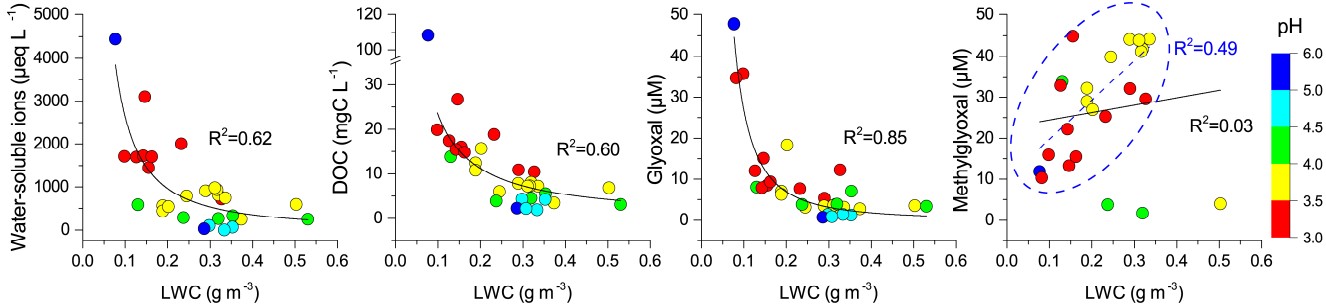

**Figure 3. Relationships of water-soluble ions, dissolved organic carbon (DOC), glyoxal and methylglyoxal with liquid water content (LWC). Color scale represents the pH range. Solid lines are empirical inverse-power fits to the data. Methylglyoxal has a good linear relationship with LWC for samples in the blue dashed circle.**

## 3.2 Gas/aqueous phases partitioning of carbonyl compounds

The simultaneous measurement of carbonyl compounds in both gas and aqueous phases enables the investigation of their partitioning between different phases (Figure 4a and Table S4). Acetone, formaldehyde, and acetaldehyde were the dominant carbonyl species (92%) measured in the gas phase during cloud events, while methylglyoxal (4%) and glyoxal (1%) were the minors. Due to high $K_H$ (and solubility) (Table S4), the dicarbonyls were found much more abundant in cloud water, with methylglyoxal and glyoxal accounting for 63% and 29% of total carbonyl species, respectively, despite their low gas-phase mixing ratios. However, diverse discrepancies were observed between the measured ($F_{me}$) and theoretical ($F_{theo}$) aqueous phase fraction of the individual carbonyl compounds. The $F_{me}/F_{theo}$ ratios for each carbonyl are plotted as a function of $K_H$ in Figure 4b. The $F_{me}$ values for carbonyls with small $K_H$ were about 1–3 orders of magnitude higher than $F_{theo}$, while for highly soluble dicarbonyls the $F_{me}/F_{theo}$ ratios approached unity, similar to the result found at Schmücke mountain in the FEBUKO study (van Pinxteren et al., 2005).

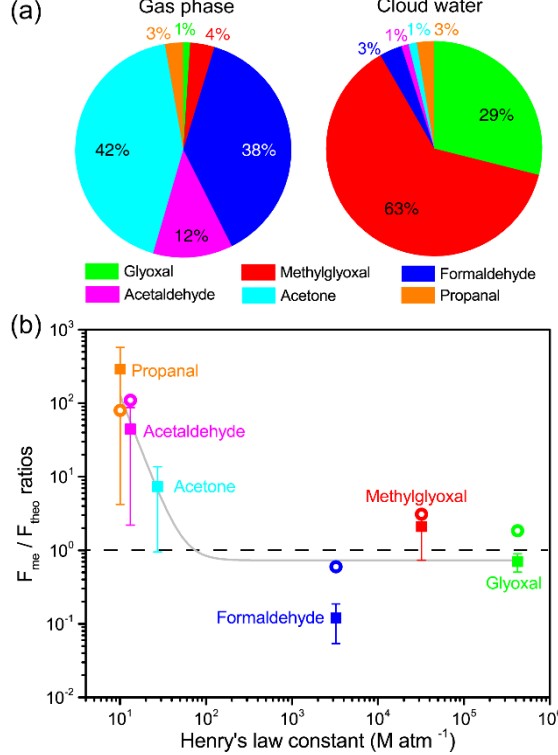

**Figure 4. (a) Mass concentration fractions of measured carbonyl compounds in gas phase and cloud water, and (b) $F_{me}/F_{theo}$ ratios as a function of Henry's law constant ($K_H$). Colored squares represent the mean $F_{me}/F_{theo}$ ratios and whiskers indicate standard deviation. For comparison, $F_{me}/F_{theo}$ ratios measured at Schmücke mountain (van Pinxteren et al., 2005) are indicated by open circles. The gray fitted line shows the decreasing trend of $F_{me}/F_{theo}$ ratios with increasing Henry's law constant for all species. The dashed line indicates the $F_{me}/F_{theo}$ ratio of 1.**

The cloudwater sulfate molality (~0.1 mol kg$^{-1}$ LWC on average) at Mt. TMS should be far from high enough to cause significant salting-in/out effect (i.e. an increased/decreased solubility of organics by higher salt concentrations) to remarkably alter the solubility of carbonyls in the dilute cloud water, although the salting-in/out effect is of particular importance in the effective uptake of carbonyl compounds by ambient particles (Waxman et al., 2015; Shen et al., 2018) where solute concentrations are high. Oligomerization on droplets surface layer induced by chemical production and adsorption has been suggested to be able to enhance the supersaturation of less-soluble carbonyls in the aqueous phase (van Pinxteren et al., 2005; Li et al., 2008). Djikaev and Tabazadeh (2003) had proposed an uptake model to account for the gas adsorption at the droplet surface, in which some adsorption parameters and adsorption isotherm need to be known. The lack of these parameters and measurement of droplet surface area or surface-to-volume in the present work did not allow us to quantify the effects of the adsorption and oligomerization. According to the simulation with some organic species (e.g., acetic acid, methanol and butanol) by Djikaev and Tabazadeh (2003), the 'overall' Henry's law constant considering both volume and surface partitioning was only <4% higher than the experimental effective Henry's law constant. Thus the adsorption and oligomerization effects may contribute to but cannot explain the observed aqueous supersaturation phenomenon here. In

contrast, using the effective Henry's law constants considering hydration for glyoxal ($4.2\times10^5$ M atm$^{-1}$) (Ip et al., 2009) and methylglyoxal ($3.2\times10^4$ M atm$^{-1}$) (Zhou and Mopper, 1990), the calculated equilibrium partitioning of these dicarbonyls in the aqueous phase is comparable to the measured fraction. Formaldehyde was deficient in the cloud water with a $F_{me}/F_{theo}$ value of 0.12, similar to the lower measured formaldehyde in the aqueous phase than that expected at equilibrium reported by Li et al. (2008). The reaction of formaldehyde with S(IV) can readily form hydroxymethanesulfonate (HMS) (Rao and Collett, 1995; Shen et al., 2012). Based on the average SO$_2$ concentration (~1 ppb) and cloud water pH (3.63) at Mt. TMS, the upper limit of in-cloud HMS formation was estimated to be 0.07 μM, which only accounts for 4.2% of total formaldehyde and thus is insufficient to explain the formaldehyde deficit.

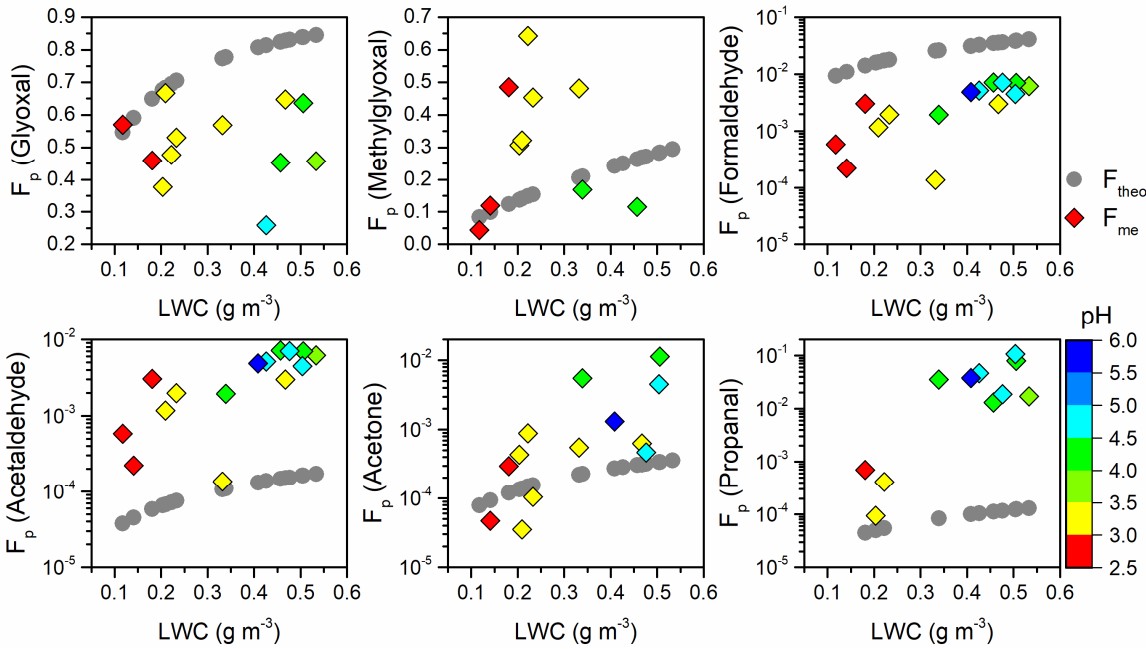

**Figure 5. Theoretical ($F_{theo}$, gray circle) and measured ($F_{me}$, colored diamond) aqueous phase fraction ($F_p$) of carbonyl compounds as a function of LWC and pH.**

Figure 5 depicts the dependence of $F_{me}/F_{theo}$ ratios of carbonyl compounds on LWC and pH. In general, the $F_{theo}$ of measured carbonyls increased to different degrees with enhanced LWC, because larger water content has a greater capacity to retain organic species. The $F_{me}$ also increased remarkably as the LWC increased, but deviated to different degrees from $F_{theo}$. For example, the $F_{me}$ values for methylglyoxal and acetone surpassed their $F_{theo}$ values when LWC exceeded ~0.2 g m$^{-3}$, whereas the $F_{me}$ values of formaldehyde and acetaldehyde were approximately parallel to their $F_{theo}$ throughout the LWC range, being one order of magnitude lower and higher, respectively. It should be noted that pH value was positively related to LWC but not involved in the $F_{theo}$ calculation, so the elevated $F_{theo}$ and increase in pH were not necessarily correlated. In contrast, the $F_{me}$ seemed to be more close to $F_{theo}$ at lower pH, but increased more rapidly than $F_{theo}$ at higher pH for the monocarbonyls except formaldehyde. For dicarbonyls, the $F_{me}$ of glyoxal slightly decreased at higher pH and showed a larger departure from $F_{theo}$,

while the $F_{me}$ of methylglyoxal far exceeded the theoretical values around pH of 3.0–3.5. Previous studies have found that the solution acidity can largely affect the reactive uptake of dicarbonyls (Gomez et al., 2015; Zhao et al., 2006). The cloud water acidity may also influence the partitioning of carbonyls, and to some extent contribute to the different gaps between $F_{me}$ and $F_{theo}$ in the present study. The complicated partitioning behaviors could be affected by both physical (e.g., interface adsorption effect) and chemical processes (e.g., chemical production from precursors) (van Pinxteren et al., 2005, and references therein). It is currently impossible to account for the results in detail. Further laboratory and theoretical studies are critically warranted.

### 3.3 Correlations between carbonyls and carboxylic acids

To investigate the potential precursors of carboxylic acids and DOM in cloud water, the correlations among all detected organic compounds and water-soluble ions were examined. Significant correlations were found for secondary water-soluble ions ($SO_4^{2-}$, $NO_3^-$ and $NH_4^+$) with glyoxal ($0.76 < r < 0.88$, $p < 0.01$) and carboxylic acids ($0.72 < r < 0.94$, $p < 0.01$). As sulfate is primarily produced by in-cloud S(IV) oxidation (Harris et al., 2013), a strong correlation ($r = 0.75$, $p < 0.01$) between oxalic acid and sulfate suggests significant in-cloud formation of oxalic acid. Likewise, the chemical cloud processing might have contributed to the secondary formation of other organic matters in the aqueous phase, such as DOM with a significant correlation with sulfate ($r = 0.83$, $p < 0.01$) (Ervens et al., 2011; Yu et al., 2005).

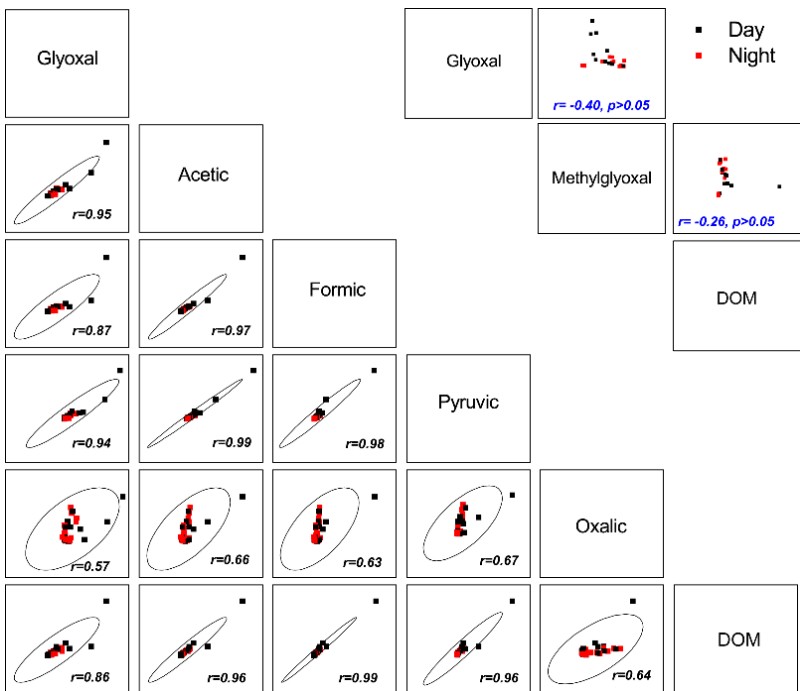

**Figure 6. Pairwise scatter plot of selected organic species in cloud water. The confidence ellipses were plotted (using OriginPro 2015 software) to indicate the strength of bivariate correlations at 99% confidence level. The confidence ellipse collapses diagonally as the correlation between two variables approaches 1 or -1, and become more circular when two variables are uncorrelated.**

Figure 6 shows the pairwise correlations (p < 0.01) among the selected organic species. Aqueous-phase glyoxal was positively correlated with all carboxylic acids (0.57 < r < 0.95) and DOM (r = 0.86) in both daytime and nighttime. Moreover, the gas-phase glyoxal exhibits positive relationships with aqueous-phase glyoxal and carboxylic acids, particularly oxalic acid (Figure S5). Many laboratory experiments have demonstrated that radical (mainly ·OH) (Lee et al., 2011; Schaefer et al., 2015) and non-radical aqueous oxidation of glyoxal (Lim et al., 2010; Lee et al., 2011; Schaefer et al., 2015; Gomez et al., 2015) can produce abundant small carboxylic acids (e.g., oxalic and formic acids), oligomers and highly oxidized organics, which subsequently lead to mass increase in SOA upon droplet evaporation (Galloway et al., 2014). In this study, the abundant methylglyoxal showed no significant correlations with glyoxal, carboxylic acids or DOM in both daytime and nighttime. Given the high solubility of glyoxal and its potential yield of carboxylic acids (Carlton et al., 2007; Lim et al., 2005; Lim et al., 2010; Blando and Turpin, 2000), glyoxal should be of great importance in the secondary organic matters formation in cloud water at Mt. TMS. In addition, as oxalic acid is predominantly formed in clouds (Myriokefalitakis et al., 2011; Ervens et al., 2011), the good interrelationships among carboxylic acids and DOM (Figure 6) indicates that carboxylic acids can contribute to DOM formation directly and/or indirectly via oxidizing to oligomers (Carlton et al., 2006; Tan et al., 2012).

## 3.4 Aqueous organics formation and cloud effects on aerosol properties

### 3.4.1 Cloud water organics formation

Cloud processing can efficiently alter the aerosol concentration and composition by nucleation and impaction scavenging (Ervens, 2015), especially at the initial stage of cloud events (Wang et al., 2011a; Li et al., 2017). On the other hand, chemical cloud processing greatly favors the in-cloud formation of sulfate (Harris et al., 2013) and SOA (Brégonzio-Rozier et al., 2016), which can remain in the particle phase upon droplet evaporation. To investigate the scavenging and changes of aerosols during cloud events, temporal variations of glyoxal, carboxylic acids, DOM and sulfate in cloud water, and ambient $PM_{2.5}$ during three cloud events (E.2, E.4, and E.5) were examined (Figure 7). Based on hourly $PM_{2.5}$ and water-soluble ions data (not shown here), the average scavenging ratios were determined to be 0.72 for $PM_{2.5}$, 0.85 for aerosol sulfate, 0.69 for nitrate and 0.68 for ammonium within the first 1–2 h of cloud processing, which were ascribed to the high cloud density, long cloud duration and little external aerosol invasion.

Figure 7 illustrates the variations of cloud water organics, sulfate and ambient $PM_{2.5}$ along with cloud evolution. Positive change rates were found during the daytime with enhanced solar radiation, while negative change rates appeared with reduced solar radiation at sunset and nighttime. This result agreed with the simulation by Huang et al. (2011), in which increasing solar radiation enhanced organic acids and SOA production through photochemical reactions. During the clean continental case E.2, the total carboxylic acids in cloud water increased by a factor of 1.9 and DOM was elevated from 3.7 to 5.7 μg m$^{-3}$ as solar radiation intensified to ~300 W m$^{-2}$, corresponding to the dramatic growth in aqueous glyoxal from 69 to 216 ng m$^{-3}$; while the increment of sulfate was relatively small, with only 0.15 μg m$^{-3}$ (i.e. 10%).

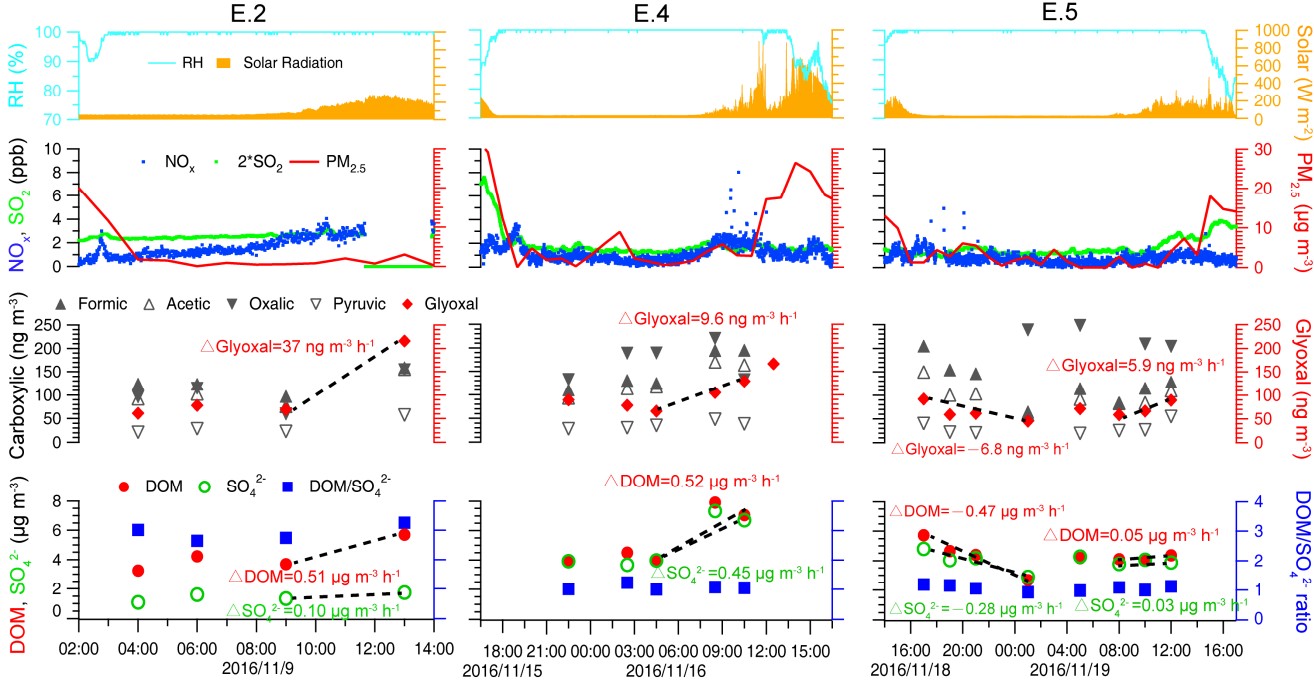

**Figure 7. Temporal variation of air equivalent concentrations of glyoxal, carboxylic acids, dissolved organic matter (DOM), $SO_4^{2-}$ and DOM/$SO_4^{2-}$ ratio in cloud water, and ambient during three cloud events (E.2, E.4 and E.5). Meteorological parameters (relative humidity and solar radiation) and air pollutants ($NO_x$, $SO_2$ and $PM_{2.5}$) are also displayed. Mass change rates of cloud water components are indicated by dashed lines and slopes.**

The oxalate/sulfate ratio can be indicative of the in-cloud oxalate formation relative to sulfate. For example, aircraft observations (Sorooshian et al., 2007; Wonaschuetz et al., 2012) have shown an increasing aerosol oxalate/sulfate ratio throughout the mixed cloud layer from 0.01 for below-cloud aerosols to 0.09 for above-cloud aerosols, suggesting more aqueous production of aerosol oxalate relative to sulfate by chemical cloud processing. Similarly, the observed oxalate/sulfate and DOM/sulfate ratios in cloud water for case E.2 increased from 0.04 to 0.09 and from 2.7 to 3.3 after sunrise, respectively, also demonstrating the increased cloudwater organics formation as contributed by cloud processing. A chamber study demonstrated that the faster photochemical uptake of glyoxal under irradiation than that in dark conditions remarkably enhanced the aqSOA formation rate by several orders of magnitude (Volkamer et al., 2009), and the radical-initiated photo-production of aqSOA mass in the daytime was predicted to be an order of magnitude higher than at nighttime (Ervens and Volkamer, 2010). Therefore, photochemical reactions are expected to enhance the cloudwater organics production (0.51 µg m$^{-3}$ h$^{-1}$) compared to sulfate (0.10 µg m$^{-3}$ h$^{-1}$) in case E.2. During the mixed E.4, the approximate growth rates of DOM (0.52 µg m$^{-3}$ h$^{-1}$) and sulfate (0.45 µg m$^{-3}$ h$^{-1}$) were comparably fast. By contrast, the growth of daytime DOM (0.05 µg m$^{-3}$ h$^{-1}$) and sulfate (0.03 µg m$^{-3}$ h$^{-1}$) during the marine E.5 was observed much slow, which could result from the relatively fewer precursors in the cleaner marine air masses. It was noted that aqueous glyoxal gradually increased after sunrise (6–37 ng m$^{-3}$ h$^{-1}$), which likely produced carboxylic acids such as oxalic acid rapidly via photo-oxidation and contributed to the formation of aqueous

organics (Carlton et al., 2007; Warneck, 2003). But at nighttime, the decrease of aqueous glyoxal (-6.8 ng m$^{-3}$ h$^{-1}$) was observed concurrent with the apparent reduction of DOM (-0.47 µg m$^{-3}$ h$^{-1}$) in E.5. Although aqueous-phase oligomers can be formed at nighttime, the oligomer formation is most likely not important in clouds (Lim et al., 2010) and thus not relevant here.

Figure 7 also shows that DOM/SO$_4^{2-}$ ratios during E.4 and E.5 remained nearly constant at ~1.0 with stronger cloud water acidity (pH of 2.96–3.68), whereas the ratios during other cloud events varied from 1.6 to 6.5 under higher pH conditions (3.62–5.94). Figure 8 shows the DOM/SO$_4^{2-}$ ratios as a function of pH values and the significantly positive relationship between DOM and sulfate, which indicates their common source of in-cloud aqueous production. The increased sulfate leads to more acidic conditions (i.e. lower pH), except the most polluted case E.1. Although the DOM also showed higher concentration in lower pH conditions, the DOM/SO$_4^{2-}$ ratios clearly decreased at lower pH range. It is well known that in-cloud oxidation of S(IV) by H$_2$O$_2$ is the predominant pathway for sulfate formation at pH < 5, within which the oxidation rate is independent of pH (Seinfeld and Pandis, 2006; Shen et al., 2012). The reduced DOM/SO$_4^{2-}$ ratios with pH suggest that DOM production was reduced compared to the sulfate in the more acidic condition. It is consistent with a previous study which found that oxalic acid production was more efficient relative to sulfate in the larger size and less acidic droplets (Sorooshian et al., 2007). Laboratory studies also found that the uptake of both glyoxal and methylglyoxal by acidic solutions increased with decreasing acid concentration, contributing to the formation of organic aerosols more efficiently (Gomez et al., 2015; Zhao et al., 2006). Additionally, there is a lack of significant competition for H$_2$O$_2$ by carbonyl compounds versus S(IV) since the reactions of carbonyls with H$_2$O$_2$ are likely negligible under cloud conditions due to their very small reactivities (Schöne and Herrmann, 2014). Although the influence mechanism of cloud water acidity on organics production remained unclear, the observed DOM/SO$_4^{2-}$ dependent trend on pH suggests that the in-cloud formation of DOM is likely more efficient under less acidic conditions.

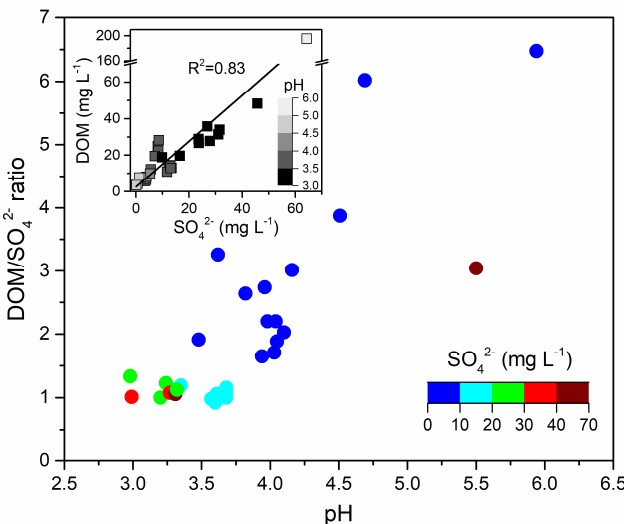

**Figure 8. DOM/SO$_4^{2-}$ ratio as a function of cloud water pH. The embedded graph shows the relationship between DOM and SO$_4^{2-}$. SO$_4^{2-}$ concentrations and pH values are both indicated by color scales.**

### 3.4.2 Impacts of cloud processing on aerosols composition and size distribution

To evaluate the impacts of cloud processing on aerosol chemistry, the major water-soluble components in pre-cloud and post-cloud aerosols (size <2.5 µm) are compared in Figure 9 and Figure S6. For the highly polluted case E.1, the total mass concentration of water-soluble components increased from 11.5 µg m$^{-3}$ in pre-cloud aerosols to 16.7 µg m$^{-3}$ in post-cloud
aerosols. A slightly elevated mass concentration of post-cloud aerosol components was also observed for mixed case E.3. Those increases in the cloud-processed aerosols were mainly contributed by sulfate and WSOM (Figure S6), suggesting the contribution of in-cloud aqueous production. In the polluted case E.1, although the mass concentration of sulfate was increased after cloud processing, the fraction slightly decreased from 57% to 51% in the post-cloud aerosols, which was mostly due to the large increases of WSOM from 19% (pre-cloud aerosols) to 28% (post-cloud aerosols). In the mixed case E.3, the fractions
of sulfate and WSOM were both enhanced in the post-cloud aerosols from 18% and 36% to 30% and 47%, respectively, with a notable increase in sodium fraction due to the influence of marine air masses. The results demonstrated the important roles of chemical cloud processing in altering aerosols composition through the aqueous sulfate and organics formation. It has been suggested that highly oxidized cloud water organics readily remain in evaporating cloud droplets and contribute to aqSOA mass production, whereas volatile products are prone to escape into the gas phase (Schurman et al., 2018). Thus we hypothesize
that the large fraction of unidentified organics in cloud water (76% in this study) could be oxidized species with low-volatility, which possibly contribute to the increased WSOM in cloud-processed aerosols. In addition, abundant interstitial particles were observed during the cloud events, similar to the previous measurements in North and South China which also observed large amounts of interstitial particles with a median diameter of 422 nm and enhanced oxalate formation in interstitial particles (Liu et al., 2018; Zhang et al., 2017). The aqueous reactions on the wetted but unactivated particles could also play important roles
in altering the particle composition and producing secondary species, like sulfate and carboxylic acids, which require more investigations in the future.

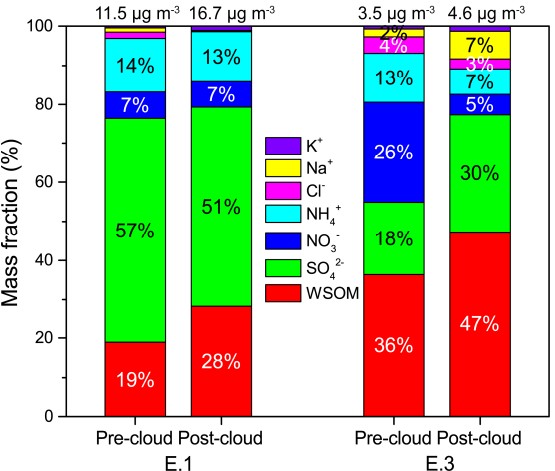

**Figure 9. Mass concentration distributions of major water-soluble components in the pre-cloud and post-cloud aerosols for cloud events E.1 and E.3.**

In Figure 10, the average mass size distributions of aerosol particles during pre-cloud and dissipation periods for cases E.1 and E.4 are compared. Multimodal distribution is apparent, with the dominant accumulation mode peaking at ~0.4 μm and a second coarse mode at ~2.0 μm. Accumulation-mode aerosols (0.1–1.0 μm) usually consist of two subgroups, the condensation and droplet modes peaking typically at 0.2–0.3 and 0.5–0.8 μm, respectively (Hinds, 2012). In this study, the overlapping of the two subgroups likely made the accumulation mode peak. For the polluted E.1, both accumulation and coarse modes aerosols decreased during cloud dissipation period compared to the pre-cloud aerosols, probably due to the cloud scavenging effect. In contrast, the concentrations of aerosols with a diameter of ~0.6–1.1 μm exceeded the pre-cloud ones, and the droplet-mode (0.5–1.0 μm) mass fraction increased notably after the cloud processing, from the pre-cloud 9% to 18% (dissipation period). For the mixed event E.4, though aerosols were scavenged in all modes, the cloud-processed aerosols still showed an increased droplet-mode mass fraction (19%) compared to the pre-cloud aerosols (15%). As droplet-mode aerosols are mainly produced from aqueous reactions, the increase in droplet-mode mass fraction after cloud dissipation may be associated with the in-cloud formation of sulfate and aqSOA (Blando and Turpin, 2000; Ervens et al., 2011). Model simulations reveal that the relative mass increase of droplet-mode aerosols after cloud processing can be up to ~100% for marine air masses with significantly accumulated sulfate and oxalate at 0.56 μm range (Ervens et al., 2018). Hence we can expect that sulfate (air equivalent concentration of 4.9 μg m$^{-3}$) and the low-volatile fraction of DOM (15.0 μg m$^{-3}$) measured in cloud water could be mostly retained in droplet-mode aerosols upon cloud evaporation. Although the mass size distributions of particle compositions were not measured in this study, the abundant droplet-mode oxalate, organic carbon and sulfate aerosols reported in Hong Kong (Bian et al., 2014; Gao et al., 2016), together with the clearly increased bulk sulfate and WSOM in the post-cloud aerosols (Figure S6), seem to support our hypothesis.

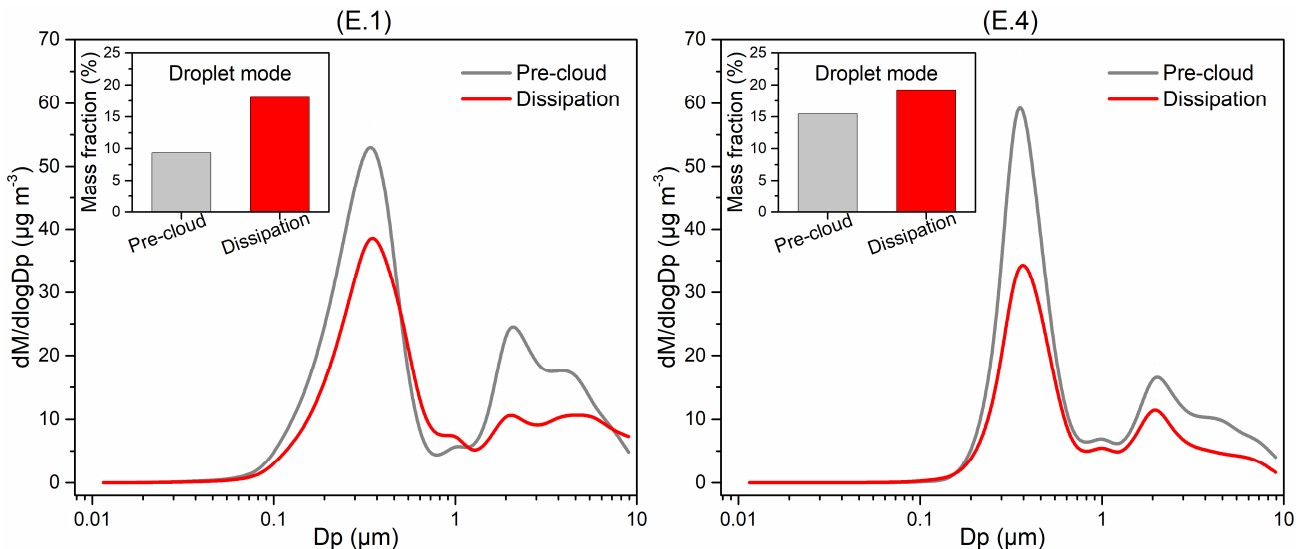

**Figure 10. Average mass size distributions of aerosol particles during the pre-cloud and dissipation periods for cloud events E.1 and E.4. Embedded graphs show the droplet-mode fraction of the total mass. A particle density of 1.65 μg m$^{-3}$ was used in this study.**

## 4 Conclusions

Gas–cloud–aerosol interactions can determine the fate of trace gases and the physicochemical properties of aerosols, but the multiphase processes in the subtropical PRD-HK region are still poorly understood. This study presents the results from a field campaign with concurrent measurements of gases, particles and cloud waters conducted at a mountain site in Hong Kong for the first time. The chemical compositions of the acidic cloud water (pH ranges of 2.96–5.94) during different cloud events were dominated by DOM and secondary inorganic ions, which were heavily influenced by anthropogenic emissions from continental air masses. Continental air masses generally contributed more pollutants to cloud water than the marine air masses did. The distinct relationships of carbonyl compounds with LWC and pH were likely controlled by their partitioning between cloud water and the gas phase. Simultaneous measurements in the two phases enabled the investigation of their partitioning behaviors. The $F_{me}$ values of dicarbonyls considering hydration reactions agreed well with their theoretical values, whereas large discrepancies were found between $F_{me}$ and $F_{theo}$ of monocarbonyls. The complicated partitioning behaviors of carbonyls possibly result from the combined physical and chemical effects, which require further investigation.

The good correlation between DOM and sulfate indicated the in-cloud formation of aqueous organics, for which abundant glyoxal likely played an important role given its significant correlations with carboxylic acids and DOM. During cloud processing, growth of cloud water DOM (0.05–0.52 μg m$^{-3}$ h$^{-1}$) and sulfate (0.03–0.45 μg m$^{-3}$ h$^{-1}$) was observed as solar radiation increased, with simultaneous increase of glyoxal (5.9–37 ng m$^{-3}$ h$^{-1}$). The cloud water DOM production seemed to be more efficient under less acidic conditions relative to sulfate. Apart from cloud scavenging of aerosol particles, cloud processing played crucial roles in changing the chemical composition and mass size distribution of particles. Remarkably increase in absolute concentrations and mass fractions of sulfate and WSOM was observed in the cloud-processed aerosols compared to the pre-cloud ones. It is expected that large amounts of sulfate and organics produced in cloud water can remain in the particle phase after cloud dissipation and lead to the mass increase in droplet-mode particles. The observations provide direct evidence for the modification of aerosols by cloud processing, promoting our understanding of the gas–cloud–aerosol interactions and multiphase chemistry of polluted coastal environments.

**Data availability**

The measurement data of cloud water chemistry that support the findings of this study are openly available in DataSpace@HKUST at https://dataspace.ust.hk/bib/KURDVK. The trace gas and PM$_{2.5}$ data used in this study are available upon request from the Hong Kong Environmental Protection Department (HKEPD) or the corresponding author (z.wang@ust.hk).

**Author contribution**

ZW, TW and YW designed the research; TL, YW, CW, YL performed the field measurement of cloud water and sample analysis; MX, CY, HY, WW conducted the measurement of trace gases and aerosols; TL, ZW, JG and HH performed data analysis and wrote the manuscript. All authors contributed to discussion and commented on the paper.

**Competing interests**

The authors declare that they have no conflict of interest.

**Acknowledgments**

The authors would like to thank Steven Poon, Bobo Wong for their support during the campaign, and to thank Hong Kong Environmental Protection Department (HKEPD) for sharing the trace gas and $PM_{2.5}$ data at Tai Mo Shan AQM station. This

work was funded by the National Key R&D Program of China (2016YFC0200503), Research Grant Council of the Hong Kong Special Administrative Region, China (25221215, 15265516, T24/504/17), and the National Natural Science Foundation of China (41605093, 41475115, 91744204). The authors greatly acknowledge the editor and anonymous reviewers for their constructive suggestions which largely improved the quality of the work.

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
