# Peer review of "Chemical characteristics of cloud water and the impacts on aerosol properties at a subtropical mountain site in Hong Kong"

_Atmospheric Chemistry and Physics, 2019_

## Referee Comment (RC1) · Anonymous Referee #2 · 4 Jun 2019

General Comments: This study reports on field measurements of trace gases, aerosols, and cloud water at a mountaintop site in Hong Kong. A valuable set of results are provided that are important for the research community interested in cloud processes, especially aqueous processing in clouds. The paper is written fairly well and the methods used seem sound. The conclusions are supported by the data. I did not find too much to comment on in terms of issues and it is my opinion that the paper was constructed well. I only have minor comments below that should be addressed prior to publication.

Specific Comments: Page 15, Line 23-26: This study showed how the oxalate:sulfate

[Figure]

ratio grows in clouds and worth noting here for the discussion:

Wonaschuetz, A., et al. (2012). Aerosol and gas re-distribution by shallow cumulus clouds: an investigation using airborne measurements, J. Geophys. Res., 117, D17202, doi:10.1029/2012JD018089.

Page 18, Line 19-21: This proposal is also supported by the following and can be added in the discussion to support the authors' speculation:

Ervens, B., et al. (2018), Is there an aerosol signature of chemical cloud processing? Atmos. Chem. Phys., 18, 16099-16119, doi: 10.5194/acp-18-16099-2018.

Technical Comments: Page 11: "organics" is spelled wrong

---

## Referee Comment (RC2) · Anonymous Referee #1 · 12 Jul 2019

The authors discuss data from cloud water and aerosol analysis at Mount Tai, Hong Kong. Their data set seems very unique not only for this location but also in general for the interpretation of sulfate and organic formation in clouds and the partitioning of organics. The manuscript is fairly well structured. However, I have numerous comments that should be addressed to clarify results and simplify the discussion. In addition, the scientific language could be improved for clarity at many places. Given the large number of my comments, I recommend a major revision.

Major comments

1) Terminology: Throughout the manuscript some ambiguous and unclear terminology

[Figure]

Is used. It should be modified accordingly:

a) Cloud water organics: A large fraction of organics dissolved in cloud water is usually comprised of small volatile compounds, such as formaldehyde, formic and acetic acid etc which likely do not contribute to aqSOA; another large fraction is often not characterized on a molecular level; e.g. (Herckes et al., 2013). Thus, sentences such as 'glyoxal is suggested to be the most likely precursor of cloud water organics' do not seem correct.

b) In-cloud aerosols: Are you referring to aerosol particles that have been scavenged and that form cloud droplets? Or do you include interstitial particles as well? In the latter, chemical reactions might be possible to (cf the large body of literature on chemical processes in wet aerosols) but time scales, products and product distributions will be likely different due to limited water amounts and much higher ionic strengths.

c) Cloud processes: I assume that you mean 'chemical reactions in cloud droplets'. 'Cloud processes' is too unspecific as it implies any physical, chemical, dynamical, meteorological . . . process related to clouds.

d) DOM (dissolved organic matter): What is 'pre-cloud DOM' and 'in-cloud DOM'? Could their loading and composition simply differ because the latter includes organics that are dissolved in cloud water whereas the former only includes low volatility and semivolatile compounds whereas highly volatile but soluble compounds are not included because they resided in the gas phase during the pre-cloud periods?

e) aqueous-phase partitioning fraction: I think it would be sufficient to call it 'aqueous phase fraction' or 'fraction that partitions to the aqueous phase'

2) Absolute vs relative numbers In several sections, mass fractions rather than absolute masses are discussed. This is sometimes misleading since, for example, a decreasing ratio might either indicate a decreasing numerator or an increasing denominator. I suggest adding absolute numbers where possible.

a) p. 8, l. 13: What are the absolute masses of Na+ and Cl- in E1 vs the marine cases? Or is the apparent increase in the fraction simply observed because other compounds (sulfate, organics etc) are less abundant?

b) p. 16, l. 15ff: This discussion is focused on the lower DOM/SO42- ratio under acidic conditions. As stated correctly, more sulfate leads to lower pH which leads to the trend of DOM/SO42- with pH as shown in Figure 8. However, I cannot follow the text in l. 19, that a lower pH leads then to decreased DOM rates. Does your data support this?

c) In Figure 9, you show clearly an increase in the DOM fraction between pre- and in-cloud aerosol. Why does the total sulfate mass apparently decrease in both cases and also organics in E.4 (sulfate from 60% x 10.7 ug/m3 -= 6.4 ug/m3 to 31% x 15.8 ug/m3 = 4.9 ug/m3 during E.1 and from 5.4 ug/m3 to 1 ug/m3 during E.4, respectively).

3) Role of pH

I do not understand why the pH is included in Figures 3 and 5. In fact, I think it is rather distracting as no explanation is given for the possible influence. As stated in the discussion if Figure 5, pH is not included in the calculation of Fp. Thus, there is no reason why F(the) should be color-coded. The relationship between LWC and pH, seems robust as it can be expected that at low LWC, droplets maybe on average smaller and thus more concentrated. However, the fact that high LWC causes high Fp and high pH should not be presented as a correlation of Fp and pH.

4) Previous studies

I appreciate the discussion of related studies. However, at many places the discussion is too brief to fully understand the previously discussed results. I suggest adding a few sentences to the places listed below.

a) p. 7, l. 7: I expect that the F/A ratio depends on potential precursors and their emission strength for formic and acetic acid. Was the study by Wang et al, 2011b performed at a similar location? Why can it be implied that formic acid is more efficiently

formed to result in a higher F/A ratio?

b) p. 7, l. 15: What are the yields of glyoxal and methylglyoxal from oxidation of their known precursors. Can such differences indeed explain the differences in the relative abundances of glyoxal, methylglyoxal and formaldehyde at the various locations? Please also add the measured values at the various locations.

c) p. 7, l. 20 – 25: Add also the numbers for the metal concentrations as measured at the various locations.

d) p. 11, l. -18 – p. 12, l. 4: This discussion is confusing and distracting. The study by Waxman et al was performed on aerosol particles where the high ionic strength is indeed sufficiently high to cause salting-out effects. The study by Shen was performed on cloud water and thus the apparent higher solubility must have different reasons as those discussed by Waxman. If the authors decide to keep this text discussing these two studies, I suggest to start with the sentence on p. 12 (cloud water molality . . .is far from high enough. . .) so that the reader knows immediately that salting-in/out does not play a role here. The last sentence (l. 4) contradicts both the previous one (salting-in/out not important in cloud water) and any reference to salting-out effects.

e) p. 11, l. 15: To what extent could oligomerization on droplet surfaces explain apparent supersaturation of less-soluble carbonyls in cloud water? If this effect indeed plays a role in the accumulation of carbonyl compounds, one might expect a stronger saturation at relatively larger drop surface areas. Can you comment on this idea? For example, do you have any measure of the drop surface area and/or drop surface-to-volume ratio in the different events? If nothing else available, one could use the aerosol particle concentration as a proxy for drop number.

f) p. 15, l. 25 ff: The relative increase of oxalate vs sulfate will depend also on the ratio of their precursors. Can you compare the SO2/VOC ratios at Mt Tai to those found by Sorooshian et al. (2007)?

**5) Partitioning**

At several places, additional reactions in the aqueous phase are discussed as possible reasons for a deviation from the theoretically calculated aqueous phase fraction. I don't understand this argument since chemical reactions convert the respective species and thus it should not be included in the measured value. Or are you referring to experimental biases due to reactions in the cloud sample after collection?

a) p. 12, l. 5: The Henry's law constants in Table S3 are the effective Henry's law constants. Thus, they include hydration.

b) p. 12, l. 9: The potential of HMs- formation could be easily estimated based on the equilibrium constants available in the literature and the measured HCHO and SO2 concentrations.

Minor comments

p. 1, l. 14: Strictly, carboxylic acids are also carbonyl compounds. Maybe specify here aldehydes and acetone.

p. 1, l. 20: 'complicated effects of both physical and chemical processes' is very vague and does not add any information here. Given that you do not identify any of these processes in the discussion, I suggest removing this fragment.

p. 2, l. 19: 'cloud-free particles' should be reworded.

p. 2, l. 29: Not only the hydrolysis of sulfate and organonitrate formation leads to differences between aerosol and cloud water O/C. Also the differences in chemical composition due to dissolution of volatile organics might add to such differences (cf. Comment 1a and 1d).

p. 2, l. 31: There are many observational studies that have shown sulfate formation in clouds in the 1980 and 1990s. At least some should be referenced here.

p. 4, l. 3: Define AIM. Did you use the AIM model for any calculation? If so, for what?

p. 5, l. 28: Here and throughout the manuscript: Numbers should be rounded to their significant digits. For example, here: 96 $\pm$ 3, instead of 96.4 $\pm$ 2.6

p. 7, l. 22: Does this sentence refer to your analysis or to the one by Harris?

p. 10, l. 7-10: The discussion of the F/A ratios is redundant here and should be included either here or on p. 7 only.

p. 10, l. 17: Are the inverse-power fits empirical or do they have any physical meaning?

p. 11, l. 8 and 10: Add 'carbonyl' here to specify the species group.

p. 14, l. 15: Add a reference for non-radical oxidation of glyoxal.

p. 15, l. 20: Direct photolysis processes are likely less efficient. 'Photolysis reactions' should be replaced by 'photochemical reactions'.

p. 16, l. 5: Oligomer formation is most likely not important in clouds and thus not relevant here, e.g. (Lim et al., 2010)

p. 16, l. 1: During E.4, the sum of sulfate and DOM formation rates (0.52 + 0.45 ug /m3/h) almost equal the total growth rate (1.07 ug/m3/h). However, during E.5 the discrepancy is much greater (0.05+ 0.03 ug/m3/h vs 0.35 ug/m3/h). Can you comment on possible reasons?

p. 16, l. 14: Is this sentence a contradiction of the previous one where you discuss a decrease in DOM?

Technical comments

p. 2, l. 18: 'contained' can be removed

p. 2, l. 19: ..was three times...

p. 4, l. 19: Text is unclear 'into 2 ml a high-pressure liquid chromatography (HPLC) grade acetonitrile'.

p. 5, l. 12: average LWC

p. 5, l. 22: at many sites

p. 7, l. 29: Three-day back trajectories

Figure 1: The red cross is very hard to see. Replace by a different color with more contracts (e.g. black) and/or increase the symbol size.

p. 10, l. 19/20: These two sentences need some grammatical corrections.

p. 10, l. 22: . . .each carbonyl compound . . .

p. 14, l. 4: Ranges should be presented as 0.72 < r < 0.94 etc

p. 15, l. 2: 'unsaturation' should be 'subsaturation'

––––––––––––––––––––––––––––––

---

## Author Comment (AC1) · 9 Sep 2019

The authors discuss data from cloud water and aerosol analysis at Mount Tai, Hong Kong. Their data set seems very unique not only for this location but also in general for the interpretation of sulfate and organic formation in clouds and the partitioning of organics. The manuscript is fairly well structured. However, I have numerous comments that should be addressed to clarify results and simplify the discussion. In addition, the scientific language could be improved for clarity at many places. Given the large number of my comments, I recommend a major revision.

**Response:** We thank the reviewer for the helpful comments on our manuscript. We have made all of the suggested changes and clarifications. The reviewer's comments are in black and our responses are in blue, and the changes in the manuscript are in *italic*.

Major comments

1) Terminology: Throughout the manuscript some ambiguous and unclear terminology is used. It should be modified accordingly:

a) Cloud water organics: A large fraction of organics dissolved in cloud water is usually comprised of small volatile compounds, such as formaldehyde, formic and acetic acid etc which likely do not contribute to aqSOA; another large fraction is often not characterized on a molecular level; e.g. (Herckes et al., 2013). Thus, sentences such as 'glyoxal is suggested to be the most likely precursor of cloud water organics' do not seem correct.

**Response:** Thanks for pointing out the issue. We agree with this comment that a large fraction of cloud water organics such as highly oxidized materials are not characterized. We have clarified this and revised the relevant parts. The revised texts are as follows,

*'The abundant glyoxal showed positive correlations with all measured carboxylic acids and DOM.'*

*'…glyoxal should be of great importance in the secondary organic matters formation in cloud water at Mt. TMS.'*

*'…for which abundant glyoxal likely played an important role given its significant correlations with carboxylic acids and DOM.'*

b) In-cloud aerosols: Are you referring to aerosol particles that have been scavenged and that form cloud droplets? Or do you include interstitial particles as well? In the latter, chemical reactions might be possible to (cf the large body of literature on chemical processes in wet aerosols) but time scales, products and product distributions will be likely different due to limited water amounts and much higher ionic strengths.

**Response:** The 'in-cloud aerosol' in the present study refers to the fine aerosols sampled during cloud event by an aerosol sampler, equipped with a 10-µm cut-size inlet and a 2.5-µm cut-size selective impactor. Since most of the cloud droplets are larger than 3 µm, the aerosols sampled during the cloud event were interstitial aerosols, together with some residual particles of smaller droplets <3 µm. To clarify, we have revised the text to include this information in the methodology and also revised the relevant terms in other parts.

*'Daily fine aerosol samples were collected on quartz filters (47 mm diameter, Pall Inc.) using a four-channel sampler (Thermo Anderson, RAAS-400, USA) with a size-selective inlet remove*

*particles/droplets larger than 2.5 µm*, with a flow rate of 16.7 L min⁻¹ and sampling duration of 23 hours. The sample filters were then refrigerated at -20 °C before laboratory analysis. An ambient ion monitor (URG 9000) *with a 2.5 µm cut-size cyclone inlet* was used to measure the hourly concentrations of water-soluble ions in $PM_{2.5}$. *During the cloud event, the collected fine aerosols were most of interstitial aerosols, together with some residual particles of smaller droplets < 3 µm.*'

'…the major water-soluble components in *pre-cloud aerosols* and *in-cloud interstitial aerosols (size < 2.5 µm)* are compared in Figure 9.'

c) Cloud processes: I assume that you mean 'chemical reactions in cloud droplets'. 'Cloud processes' is too unspecific as it implies any physical, chemical, dynamical, meteorological …process related to clouds.

**Response:** We have revised the text to clarify this. The '*cloud processing*' is used when we discuss the combined effects of physical and chemical processes, and '*chemical cloud processing*' is used to refer to the chemical reactions in cloud droplets.

d) DOM (dissolved organic matter): What is 'pre-cloud DOM' and 'in-cloud DOM'? Could their loading and composition simply differ because the latter includes organics that are dissolved in cloud water whereas the former only includes low volatility and semivolatile compounds whereas highly volatile but soluble compounds are not included because they resided in the gas phase during the pre-cloud periods?

**Response:** We are sorry for the misleading statement. We think the relevant discussion that the reviewer mentioned is in Section 3.4.2. As described in previous comment, we have clarified the relevant terms, the DOM now refer to the dissolved organic matter in cloud waters, while 'water-soluble organic matter (WSOM)' is used to represent the OM measured in aerosols. Many previous studies have indicated the significant changes in chemical composition of aerosol organics by cloud processing. In this study, due to the lack of speciation information in aerosol organic composition, we can only compare the differences of WSOM in bulk concentrations and mass fractions between pre-cloud aerosols and in-cloud interstitial aerosols to examine the apparent effects of cloud processing on aerosol organics.

The revised text reads,

'Meanwhile, the mass fraction of *WSOM in aerosols* was elevated from 20% (pre-cloud) to 30% (*in-cloud interstitial*), and nitrate increased from 4% to 19%, probably due to the large increase in $NO_2$ (over 4-fold). For the mixed case E.4, *WSOM mass fraction in the interstitial aerosols* was twice of that in pre-cloud aerosols, consistent with the increasing trend of DOM in cloud water shown in Figure 7.'

e) aqueous-phase partitioning fraction: I think it would be sufficient to call it 'aqueous phase fraction' or 'fraction that partitions to the aqueous phase'

**Response**: Agree. 'Aqueous-phase partitioning fraction' was replaced by the more concise '*aqueous phase fraction*' in the revised manuscript.

2) Absolute vs relative numbers. In several sections, mass fractions rather than absolute masses are discussed. This is sometimes misleading since, for example, a decreasing ratio might either indicate a decreasing numerator or an increasing denominator. I suggest adding absolute numbers where possible.

**Response**: Thanks for the suggestion. Some absolute numbers have been added as suggested.

a) p. 8, l. 13: What are the absolute masses of Na+ and Cl- in E1 vs the marine cases? Or is the apparent increase in the fraction simply observed because other compounds (sulfate, organics etc) are less abundant?

**Response**: It should be noted that the special cloud event 1 was highly polluted by continental sources and anthropogenic emissions during the passage of a cold front, so the absolute concentrations of most cloud water components in E.1 were the highest and the LWC was lowest at 0.08 g m$^{-3}$. In contrast, the clean marine E.6 with the highest LWC of 0.35 g m$^{-3}$ generally had the lowest solute concentrations. The figure below compares the absolute concentrations of major components in cloud water among different events, and has been added as Figure S2 in the revised SI.

The absolute mass concentrations of Na$^+$ and Cl$^-$ in cloud water are 3.0 and 10 mg L$^{-1}$ for continental polluted E.1, respectively, which are much more abundant than the Na$^+$ (0.2 mg L$^{-1}$) and Cl$^-$ (1.0 mg L$^{-1}$) in cloud water for continental E.2. However, the mass fractions of Na$^+$ and Cl$^-$ between E.1 and E.2 are comparable, with 1% for Na$^+$ and 3% for Cl$^-$ in E.1 vs. 0.4% and 2% in E.2. So the variation of absolute concentrations of other major compounds seems to have little effect on mass fractions of Na$^+$ and Cl$^-$, possibly because these main components were contributed by similar (continental) sources and changed synchronously. In comparison, the concentrations of Na$^+$ and Cl$^-$ in cloud water in marine E.5 were 2.5 and 5.9 mg L$^{-1}$, accounting for 5% and 11% of total mass, respectively. Their fractions are close to those for other marine-related cloud events, although the absolute concentrations also varied.

We have added the absolute mass concentration and revised the text, as follows,

'The concentration and distributions of major components in cloud water during six cloud events are compared in Figure 2a *and Figure S2*.'

'For example, continental E.1, *which was heavily polluted by anthropogenic emissions within the passage of a cold front (Table S2 and Figure S2)*, exhibited the largest amount of major components (393.9 mg L$^{-1}$) whereas marine E.6 had the least (15.7 mg L$^{-1}$).'

'Influenced by marine air masses, *the concentration (and proportions) of Cl$^-$ and Na$^+$ notably increased from 0.2 mg L$^{-1}$(0.4%) and 1.0 mg L$^{-1}$ (2%) in continental cloud water (E.2) to 2.5 mg L$^{-1}$ (5%) and 5.9 mg L$^{-1}$ (11%) in the marine one (E.5), respectively.'*

[Figure]

**Figure S2.** *Absolute mass concentrations of major components in cloud water for case E.1-6.*

b) p. 16, l. 15ff: This discussion is focused on the lower DOM/SO42- ratio under acidic conditions. As stated correctly, more sulfate leads to lower pH which leads to the trend of DOM/SO42- with pH as shown in Figure 8. However, I cannot follow the text in l. 19, that a lower pH leads then to decreased DOM rates. Does your data support this?

**Response**: Yes, as the reviewer stated, more sulfate leads to lower pH, but it doesn't necessarily lead to the lower DOM/SO$_4^{2-}$ ratio at lower pH. As the DOM and sulfate in cloud water showed good correlation because of their common in-cloud aqueous production, the DOM concentrations in the high sulfate and lower pH cases were also much higher than the higher pH cases. It is well known that the in-cloud oxidation of S(IV) by H$_2$O$_2$ is the predominant pathway for sulfate formation at pH less than ~5, within which the oxidation rate is independent of pH. If the DOM aqueous formation is also independent of acidity, the DOM/SO$_4^{2-}$ ratio would also show no clear dependence on pH. However, the observation showed that the DOM/sulfate ratios clearly decreased at lower pH, implying less DOM production compared to sulfate in the lower pH condition. We have revised the Figure 8 and the text to make it clear.

[Figure]

***Figure 8.*** *DOM/SO$_4^{2-}$ ratio as a function of cloud water pH. The embedded graph shows the relationship between DOM and SO$_4^{2-}$. SO$_4^{2-}$ concentrations and pH values are both indicated by color scales.*

'Figure 8 shows the DOM/SO$_4^{2-}$ ratio as a function of pH values and the significantly positive relationship between DOM and sulfate, *which indicates their common source of in-cloud aqueous production. The increased sulfate leads to more acidic conditions (i.e. lower pH), except the most polluted case E.1. Although the DOM also showed higher concentration in lower pH condition, the DOM/SO$_4^{2-}$ ratios clearly decreased at lower pH range. It is well known that the in-cloud oxidation of S(IV) by H$_2$O$_2$ is the predominant pathway for sulfate formation at pH<5, within which the oxidation rate is independent of pH (Seinfeld and Pandis, 2006; Shen et al., 2012).The reduced DOM/SO$_4^{2-}$ ratios with pH suggest that DOM production was reduced compared to the sulfate in the more acidic condition. It is consistent with a previous study which found that* oxalic acid production was more efficient relative to sulfate in the larger size and less acidic droplets (Sorooshian et al., 2007). Laboratory studies also found that the uptake of both glyoxal and methylglyoxal by acidic solutions increased with decreasing acid concentration, contributing to the formation of organic aerosols more efficiently *(Gomez et al., 2015; Zhao et al., 2006). Additionally, the possibility of competition for H$_2$O$_2$ between carbonyl*

*compounds and S(IV) can be excluded because substantial $H_2O_2$ is usually found in cloud water (Shen et al., 2012). Although the influence mechanism of cloud water acidity on organics production remained unclear, the observed $DOM/SO_4^{2-}$ dependent trend on pH suggests that the in-cloud formation of DOM is likely more efficient under less acidic conditions.'*

c) In Figure 9, you show clearly an increase in the DOM fraction between pre- and in-cloud aerosol. Why does the total sulfate mass apparently decrease in both cases and also organics in E.4 (sulfate from 60% x 10.7 ug/m3 -= 6.4 ug/m3 to 31% x 15.8 ug/m3 = 4.9 ug/m3 during E.1 and from 5.4 ug/m3 to 1 ug/m3 during E.4, respectively).

**Response:** We are sorry for the unclear terminology that may mislead the reviewer on this issue. As we discussed in the previous response (#1b), the in-cloud aerosol here we discussed in Fig 9 was the interstitial aerosols. Because of the cloud scavenging effects, many species in the pre-cloud aerosols were scavenged, especially the sulfate. Therefore, the total sulfate mass decreased in both cases, and the organics decreased in E.4. The highly polluted E.1 case, which showed increase in mass concentrations of aerosol DOM, nitrate, and ammonium, was exceptional. Due to the short cloud duration (~2 h) and low LWC (0.08 g m$^{-3}$) of cloud event E.1, the cloud scavenging of aerosol particles was ineffective. On the other hand, the intrusion of much-elevated air pollutants (including $NO_x$, $SO_2$, aged aerosols, etc.) carried by cold front passage (Figure S1 and Table S2) could lead to an increase in the mass of aerosol compositions.

We have updated the Fig 9 to make the terminology clearer and also included a figure with absolute mass concentrations in SI.

[Figure]

*Figure S6. Absolute mass concentration of major water-soluble components in the pre-cloud aerosols and in-cloud interstitial aerosols for cloud events E.1 and E.4.*

3) Role of pH

I do not understand why the pH is included in Figures 3 and 5. In fact, I think it is rather distracting as no explanation is given for the possible influence. As stated in the discussion in Figure 5, pH is not included in the calculation of Fp. Thus, there is no reason why F(the) should be color-coded. The relationship between LWC and pH, seems robust as it can be expected that at low LWC, droplets maybe

on average smaller and thus more concentrated. However, the fact that high LWC causes high Fp and high pH should not be presented as a correlation of Fp and pH.

**Response:** As previous studies suggested, the phase partitioning and chemical reactions were both affected by the pH and LWC (Tilgner et al., 2005). In cloud droplets, acidity is generally related to soluble ions and can affect the solute composition in many ways, e.g., promoting the dissolution of trace metals, affecting uptake of trace gases and formation of aqueous organics (Cini et al., 2002; Deguillaume et al., 2005; Benedict et al., 2012; Sorooshian et al., 2007; Gomez et al., 2015; Straub, 2017). In Figure 3b, we intended to show the particular relationships between individual carbonyls and pH, which are quite different from the inverse-power fits for soluble ions, DOM, carboxylic acids and trace metals, and try to link the discussion in the respect of aqueous partitioning of carbonyls in Section 3.2. To make it clearer, we have revised the Fig 3 by moving the Fig 3b to SI, and include the pH dependence pattern as colored in Fig 3, as shown below.

[Figure]

*Figure 3. Relationships of water-soluble ions, dissolved organic carbon (DOC), glyoxal and methylglyoxal with liquid water content (LWC). Color scale represents the pH range. Solid lines are empirical inverse-power fits to the data. Methylglyoxal has better linear fitting curves for samples within blue dashed circles.*

The text was also revised as follows,

'Similar inverse-power relationships of water-soluble ions, DOC and carboxylic acids with pH were also found *(Figure 3 and Figure S4)*. Increased air pollution and secondary acid ions formation likely made the cloud water more acidic, in turn promoting the dissolution of trace metals (Li et al., 2017). *Unexpectedly, individual carbonyl compounds showed different relationships with LWC and pH. For instance, as LWC and pH increased, glyoxal concentrations decreased in a power function while methylglyoxal tended to increase linearly.*'

Regarding Fig 5, we agree that the theoretical partitioning fraction $F_{theo}$ should not be colored. The $F_{theo}$ was updated by gray circle in the revised manuscript.

It is true that smaller droplets with low LWC are usually more concentrated and acidic. However, the pH is fundamentally determined by the relative amounts of anions and cations rather than the absolute total amounts of solutes. Although $F_{me}$ generally increases with higher LWC in Figure 5, the gaps between $F_{me}$ and $F_{theo}$ vary differently for individual carbonyls, suggesting influences from unknown factors. As laboratory studies show that solution acidity is able to affect the reactive uptake of dicarbonyls (Gomez et al., 2015; Zhao et al., 2006), the cloud water acidity could be a possible factor influencing the actual partitioning of carbonyls, leading to the largely different gaps between $F_{me}$ and $F_{theo}$. So we think that it is better to mark pH values to imply the possible effects of acidity on actual aqueous partitioning of carbonyls in addition to LWC, though the mechanism is currently unclear.

The text is also revised, as follows,

'For dicarbonyls, the $F_{me}$ of glyoxal slightly decreased at higher pH and showed a larger departure from $F_{theo}$, while the $F_{me}$ of methylglyoxal far exceeded the theoretical values around pH of 3.0–3.5. *Previous studies have found that the solution acidity can largely affect the reactive uptake of dicarbonyls (Gomez et al., 2015; Zhao et al., 2006). The cloud water acidity may also influence the partitioning of carbonyls, and to some extent contribute to the different gaps between $F_{me}$ and $F_{theo}$ in the present study.*'

4) Previous studies

I appreciate the discussion of related studies. However, at many places the discussion is too brief to fully understand the previously discussed results. I suggest adding a few sentences to the places listed below.

**Response:**  As suggested, the discussions have been revised or added.

a) p. 7, l. 7: I expect that the F/A ratio depends on potential precursors and their emission strength for formic and acetic acid. Was the study by Wang et al, 2011b performed at a similar location? Why can it be implied that formic acid is more efficiently formed to result in a higher F/A ratio?

**Response:** Carboxylic acids in the atmosphere mainly come from direct emissions (e.g., anthropogenic sources and biomass burning) and secondary formation through photochemical reactions in the gas phase. The F/A ratio was first proposed as a marker of the relative importance of these two sources for carboxylic acids in the gas phase (Talbot et al., 1988), and then was extended to the rainwater (Fornaro and Gutz, 2003) and cloudwater (Wang et al., 2011). Direct anthropogenic emission of acetic acid by vehicles is higher than of formic acid, which results in F/A ratios much less than 1.0, whereas photochemical oxidation of natural hydrocarbons in remote areas likely leads to higher concentrations of formic acid than acetic acid and thus the increase in F/A ratios (> 1.0) (Talbot et al., 1988; Fornaro and Gutz, 2003). Therefore, the F/A ratio is a robust and feasible method to infer the dominant sources of carboxylic acids and has been used in those previous studies. Unlike the gas phase, F/A ratio in the liquid phase (rainwater or cloudwater) is expected to be higher than that in the gas phase at equilibrium conditions, and is dictated by Henry's law constants, dissociation constants of formic and acetic acids and pH. The F/A ratio in the gas phase can be calculated from the corresponding liquid phase, and can be used for roughly evaluating the dominant sources.

The study by Wang et al, 2011b was performed at Mt. Heng (1269 m) in south-central China, which is about 550 km north to the coastal Mt. Tai Mo Shan in Hong Kong. Both sites represent the regional atmospheric background environment surrounded by vegetation. Therefore, we think the F/A ratio method is applicable to this study.

We have revised manuscript by adding more discussions and relocating the relevant discussion to Section 3.1.2 on Page 10, according to this and below comments.

'*The formic-to-acetic acid (F/A) ratio has been suggested to be a useful indicator of sources of carboxylic acids from direct emissions (e.g., anthropogenic sources, biomass burning) or secondary photochemical formation, in the gas phase (Talbot et al., 1988), rainwater (Fornaro and Gutz, 2003) and cloudwater (Wang et al., 2011b). Direct anthropogenic emission of acetic acid from vehicle-related sources is higher than of formic acid, resulting in F/A ratios much less than 1.0, whereas photochemical oxidation of natural hydrocarbons leads to higher concentrations of formic acid than acetic acid, and therefore the increase in F/A ratios (> 1.0) (Talbot et al., 1988; Fornaro and Gutz, 2003). The F/A ratio in the liquid phase (rainwater or cloudwater) is expected to be higher than in the gas phase at equilibrium conditions, which is dictated by Henry's law constants, dissociation constants of formic*

*and acetic acids and pH. So the corresponding gas-phase F/A ratio can be calculated from the aqueous concentrations to evaluate the dominant sources.* In this study, a remarkable correlation between formic and acetic acid (r = 0.97, p < 0.01) suggests their similar sources or formation pathways. The high F/A ratios (1.2–1.9) than 1.0 for E.2–5 (Table S2) indicates the more important secondary formation for carboxylic acids in cloud water. In contrast, the F/A ratios in E.1 and E.6 were 0.4 and 0.5, respectively, suggesting the significant contributions from direct emissions during these two events. In addition, despite the decrease of total concentrations, the proportion of oxalic acid notably increased from 5% to 58% under more influence of marine air masses.'

b) p. 7, l. 15: What are the yields of glyoxal and methylglyoxal from oxidation of their known precursors. Can such differences indeed explain the differences in the relative abundances of glyoxal, methylglyoxal and formaldehyde at the various locations? Please also add the measured values at the various locations.

**Response:** The measured values at previous studies were added in Table S1, and more discussions were added in the revised text. According to Ervens et al. (2011) and Nishino et al. (2010), the glyoxal and methylglyoxal yields from isoprene at high $NO_x$ level were round 0.05 and 0.20, respectively. While the two yields from toluene are approximately equal (0.14 and 0.12, respectively, at high $NO_x$ level), methylglyoxal yields from xylene exceed the ones of glyoxal by a factor of 5 (0.08 and 0.47 respectively) (Nishino et al., 2010). Glyoxal and methylglyoxal are not only first, but also second-generation products in the oxidation of isoprene and, thus, their concentration ratio changes over time depending on the availability of oxidants. In general, the overall yields of these aldehydes from isoprene are much more uncertain, but are expected much lower than those from the oxidation of aromatics (Ervens et al., 2013), and the latter also contributes to higher yields of methylglyoxal than glyoxal. The high aromatics concentration at Mt. TMS (Table S1) and different precursors ratios compared to other studies, likely lead to the observed different pattern of methylglyoxal and glyoxal.

Table S1 below lists the measured values at various locations and is added in the SI.

**Table S1.** Comparison of glyoxal and methylglyoxal in cloud water [µM] and their gas-phase precursors [ppb] as well as pollutants concentrations [ppb] at Mt. TMS and other sites.

| | Glyoxal | Methylglyoxal | Isoprene | Benzene | Toluene | Xylene | $NO_x$ | $O_3$ | References |
|---|---|---|---|---|---|---|---|---|---|
| Mt. TMS | 6.7 | 19.1 | 0.16 | 0.5 | 2.3 | 0.9 | 3 | 31 | this study |
| Whistler, Canada | 0.6-1.8 | 0.5-7.4 | 0.6 | 0.05 | 0.1 | | 4 | 25 | (Ervens et al., 2013) |
| Davis, USA | 1.3-8.7 | 0.1-0.9 | 0.2 | 2 | 4 | | 20 | 70 | (Ervens et al., 2013) |
| Mt. Schmücke, Germany | 0.8-11.3 | 0.4-3.3 | | | | | | | (van Pinxteren et al., 2005) |
| Puy de Dôme, France | 0.13-0.89 | 0.01-0.22 | | up to 1.1[a] | | | | | (Deguillaume et al., 2014; Barbet et al., 2016) |

[a] sum of observed benzene, toluene and ethylbenzene (Barbet et al., 2016)

The text is also revised, as follows,

*'The nearly triple abundance of methylglyoxal than glyoxal at Mt. TMS differed from the previous observations at Puy de Dôme, France (Deguillaume et al., 2014), Mt. Schmücke, Germany (van Pinxteren et al., 2005), and Davis, USA (Ervens et al., 2013), where glyoxal concentrations were 2 to 10 times higher than methylglyoxal, but was similar to the results observed at Whistler, Canada (Ervens*

*et al., 2013) where the methylglyoxal/glyoxal ratio was much higher. These different patterns could partially be attributed to the large differences in precursors at various locations (Table S1) and also the availability of oxidants. Generally, the overall yields of these aldehydes from isoprene are much lower than those from the oxidation of aromatics (Ervens et al., 2013), and the latter also contributes to higher yields of methylglyoxal than glyoxal (Ervens et al., 2011). For example, the glyoxal and methylglyoxal yields from toluene are approximately equal (0.14 and 0.12, respectively, at high $NO_x$ level), and methylglyoxal yields from xylene exceed the ones of glyoxal by a factor of 5 (0.08 and 0.47, respectively) (Nishino et al., 2010; Ervens et al., 2011). The higher aromatics concentrations (toluene of 2.3 ppb, xylene of 0.9 ppb) than the biogenic isoprene (0.16 ppb) measured at Mt. TMS are expected to be the important precursors of these aldehydes and lead to the different ratio observed in the cloud water.*

c) p. 7, l. 20 – 25: Add also the numbers for the metal concentrations as measured at the various locations.

**Response:** Added as suggested, as shown below,

'Aluminium (131.9 μg $L^{-1}$) dominated the cloud water trace metals, of which the concentration was comparable to that measured at other mountain sites in China *(99.7 to 157.3 μg $L^{-1}$)* (Li et al., 2017). *Transition metals of Fe, Cu and Mn, which play important roles in the heterogeneous catalytical formation of sulfate (Harris et al., 2013), were also found to be abundant in the cloud water, with mean concentrations of 50.6, 10.0 and 5.9 μg $L^{-1}$, respectively.* The toxic Pb concentration in cloud water (18.7 μg $L^{-1}$) was tens of times higher than *that observed at sites in Europe (1.4 μg $L^{-1}$) (Fomba et al., 2015) and America (0.6 μg $L^{-1}$)* (Straub et al., 2012), probably due to traffic emissions from the surrounding city-cluster.'

d) p. 11, l. -18 – p. 12, l. 4: This discussion is confusing and distracting. The study by Waxman et al was performed on aerosol particles where the high ionic strength is indeed sufficiently high to cause salting-out effects. The study by Shen was performed on cloud water and thus the apparent higher solubility must have different reasons as those discussed by Waxman. If the authors decide to keep this text discussing these two studies, I suggest to start with the sentence on p. 12 (cloud water molality… is far from high enough…) so that the reader knows immediately that salting-in/out does not play a role here. The last sentence (l. 4) contradicts both the previous one (salting in/out not important in cloud water) and any reference to salting-out effects.

**Response:** It should be noted that the study by Shen et al. (2018) was performed on ambient aerosol particles in urban Beijing, China, rather than on cloud water. As suggested, we simplified these sentences and started immediately with the statement that salting effects were not important in cloud water. The revised text is as follows:

*'The cloudwater sulfate molality (~0.1 mol $kg^{-1}$ LWC on average) at Mt. TMS should be far from high enough to cause significant salting-in/out effect (i.e. an increased/decreased solubility of organics by higher salt concentrations) to remarkably alter the solubility of carbonyls in the dilute cloud water,* although the salting-in/out effect is of particular importance in the effective uptake of carbonyl compounds by concentrated solutions (Waxman et al., 2015) *and ambient particles (Shen et al., 2018).*'

e) p. 11, l. 15: To what extent could oligomerization on droplet surfaces explain apparent supersaturation of less-soluble carbonyls in cloud water? If this effect indeed plays a role in the accumulation of carbonyl compounds, one might expect a stronger saturation at relatively larger drop

surface areas. Can you comment on this idea? For example, do you have any measure of the drop surface area and/or drop surface-to-volume ratio in the different events? If nothing else available, one could use the aerosol particle concentration as a proxy for drop number.

**Response**: As the literature suggested (e.g., Li et al., 2008), the oligomerization on droplet surface may not occur due to the low solubility for these carbonyls (formaldehyde, acetaldehyde and acetone). Djikaev and Tabazadeh (2003) had proposed an uptake model to account for the gas adsorption at the droplet surface, in which several adsorption parameters (e.g., $\Gamma$, b and K) are needed to perform the calculation. However, these parameters for the species studied in the present work were not well quantified, and the drop surface area or drop surface-to-volume was not measured in the present work. Thus it is impossible to perform a quantitative estimation on the oligomerization and adsorption effects. According to simulations with some example organic species (e.g., acetic acid, methanol and butanol) by Djikaev and Tabazadeh (2003), the 'overall' Henry's law constant considering both volume and surface partitioning was only <4% higher than the experimental effective Henry's law constant. Therefore, we think the adsorption and oligomerization effects may contribute to but cannot explain the observed aqueous supersaturation phenomenon here. We have revised the text by adding more discussion, as follows,

'Oligomerization on droplets surface layer induced by chemical production and adsorption *has been suggested to be able to enhance the supersaturation of less-soluble carbonyls in the aqueous phase (van Pinxteren et al., 2005;Li et al., 2008). Djikaev and Tabazadeh (2003) had proposed an uptake model to account for the gas adsorption at the droplet surface, in which some adsorption parameters and adsorption isotherm need to be known. The lack of these parameters and measurement of droplet surface area or surface-to-volume in the present work did not allow us to quantify the effects of the adsorption and oligomerization. According to the simulation with some organic species (e.g., acetic acid, methanol and butanol) by Djikaev and Tabazadeh (2003), the 'overall' Henry's law constant considering both volume and surface partitioning was only <4% higher than the experimental effective Henry's law constant. Thus the adsorption and oligomerization effects may contribute to but cannot explain the observed aqueous supersaturation phenomenon here.*'

f) p. 15, l. 25 ff: The relative increase of oxalate vs sulfate will depend also on the ratio of their precursors. Can you compare the SO2/VOC ratios at Mt Tai to those found by Sorooshian et al. (2007)?

**Response:** As the reviewer pointed out that the relative increase of oxalate vs sulfate should depend on their precursors, it is not feasible to compare the absolute oxalate/sulfate ratio among different locations because the absolute $SO_2$ and VOC concentration and even $SO_2$/VOC ratios can be quite different. We shall make it clear that the use of oxalate/sulfate ratio change here is to compare the oxalate formation relative to sulfate at different stages of one cloud event or in aerosols with and without cloud processing, rather than to compare the absolute oxalate/sulfate ratios in aerosols or cloud water at different locations. Besides, the oxalate/sulfate ratios studied are for cloud water at Mt. TMS, but were for aerosols in Sorooshian et al. (2007), and we won't compare the ratio directly. Moreover, the oxalate production does not exhibit the same degree of sensitivity to its precursor concentrations as does sulfate to $SO_2$, since the multistep formation of oxalate may not be in direct proportion to its precursor VOC (Sorooshian et al., 2006). The average concentrations of $SO_2$, toluene, ethane and isoprene were 0.8, 2.3, 1.8 and 0.16 ppbv in this study, respectively, in comparison to those of 0.5, 0.06, 0.05 and 0.04 ppb in a flight measurement over northeast America (Sorooshian et al., 2006). No clear conclusions on oxalate/sulfate ratio can be simply drawn from the comparison.

In the revised manuscript, we revised the text with more discussions, as follows:

*'The oxalate/sulfate ratio can be indicative of the in-cloud oxalate formation relative to sulfate. For example, aircraft observations (Sorooshian et al., 2007; Wonaschuetz et al., 2012) have shown an increasing aerosol oxalate/sulfate ratio throughout the mixed cloud layer from 0.01 for below-cloud aerosols to 0.09 for above-cloud aerosols, suggesting more aqueous production of aerosol oxalate relative to sulfate by cloud processing. Similarly, the observed oxalate/sulfate and DOM/sulfate ratios in cloud water for case E.2 increased from 0.04 to 0.09 and from 2.7 to 3.3 after sunrise, respectively, also demonstrating the increased cloudwater organics formation as contributed by cloud processing'.*

5) Partitioning

At several places, additional reactions in the aqueous phase are discussed as possible reasons for a deviation from the theoretically calculated aqueous phase fraction. I don't understand this argument since chemical reactions convert the respective species and thus it should not be included in the measured value. Or are you referring to experimental biases due to reactions in the cloud sample after collection?

**Response:** In the theoretical calculation, only the processes represented in effective Henry's law partitioning was considered. As stated by the reviewer, the chemical reactions may convert the respective species, resulting less aqueous concentration than the equilibrium concentration corresponding to the gas phase. Some fast chemical reactions producing less-soluble organics or consuming carbonyls that may contribute to the supersaturation or undersaturation, respectively, were not considered in the theoretical calculation and could be a partial reason for the deviation of the measurement from the theoretical fraction. We clarify the relevant part in the revised text,

'The complicated partitioning behaviors could be affected by both physical (e.g., interface adsorption effect) and chemical processes (e.g., *fast aqueous reactions producing less-soluble organics and/or consuming dicarbonyls that result in a disequilibrium between gas and aqueous phases*) (van Pinxteren et al., 2005, and references therein). It is currently impossible to account for the results in detail. Further laboratory and theoretical studies are critically warranted.'

'a) p. 12, l. 5: The Henry's law constants in Table S3 are the effective Henry's law constants. Thus, they include hydration.

**Response**: Thanks for the correction. The text was revised as follows:

*'Using the effective Henry's law constants considering hydration for glyoxal ($4.2 \times 10^5$ M atm$^{-1}$) (Ip et al., 2009) and methylglyoxal ($3.2 \times 10^4$ M atm$^{-1}$) (Zhou and Mopper, 1990), the calculated equilibrium partitioning of these dicarbonyls in the aqueous phase is comparable to the measured fraction. In contrast, the less-soluble mono-carbonyls are still supersaturated'*

b) p. 12, l. 9: The potential of HMs- formation could be easily estimated based on the equilibrium constants available in the literature and the measured HCHO and SO2 concentrations.

**Response**: Thanks for the helpful suggestion. We estimated the potential of hydroxymethanesulfonate (HMS) formation from the literature data and found that it only has a minor contribution to HCHO deficit. More discussions are added in the revised text, as follows,

'Formaldehyde was deficient in the cloud water, with a $F_{me}/F_{theo}$ value of 0.12. *It is similar to the lower measured formaldehyde in aqueous phase than that expected at equilibrium reported by Li et al. (2008), who suggested that it was probably associated with aqueous oxidation of formaldehyde. The reaction*

*of formaldehyde with S(IV) can readily form hydroxymethanesulfonate (HMS) (Rao and Collett, 1995; Shen et al., 2012). Based on the average SO$_2$ concentration (~1 ppb) and cloud water pH (3.63) at Mt. TMS, the upper limit of in-cloud HMS formation was estimated to be 0.07 µM, which only accounts for 4.2% of total formaldehyde and thus is insufficient to explain the formaldehyde deficit.'*

5    Minor comments

p. 1, l. 14: Strictly, carboxylic acids are also carbonyl compounds. Maybe specify here aldehydes and acetone.

**Response**: At the first appearance of 'carbonyl compounds' in Abstract and main text, we specified it as aldehydes and acetone, i.e. 'carbonyl compounds (*refer to aldehydes and acetones*)'.

10    p. 1, l. 20: 'complicated effects of both physical and chemical processes' is very vague and does not add any information here. Given that you do not identify any of these processes in the discussion, I suggest removing this fragment.

**Response:** Accept. As the partitioning behaviors of carbonyls were not well accounted for at present, we deleted this vague conclusion.

15    p. 2, l. 19: 'cloud-free particles' should be reworded.

**Response:** It refers to ambient particles sampled during the periods without cloud events. We reworded it to be 'ambient (cloud-free) particles', the original expression used in the reference (Zhang et al., 2017).

Zhang, G., Lin, Q., Peng, L., Yang, Y., Fu, Y., Bi, X., Li, M., Chen, D., Chen, J., Cai, Z., Wang, X., Peng, P., amp, apos, an, Sheng, G., and Zhou, Z.: Insight into the in-cloud formation of oxalate based
20    on in situ measurement by single particle mass spectrometry, Atmospheric Chemistry and Physics, 17, 13891-13901, 10.5194/acp-17-13891-2017, 2017.

p. 2, l. 29: Not only the hydrolysis of sulfate and organonitrate formation leads to differences between aerosol and cloud water O/C. Also the differences in chemical composition due to dissolution of volatile organics might add to such differences (cf. Comment 1a and 1d).

25    **Response**: We agree that many other factors such as the dissolution of volatile organics might add to the O/C differences. However, in this sentence, we discussed the differences in organic molecular compositions between cloud water and ambient particles under similar O/C ratio condition as investigated by Boone et al., (2015). This sentence was clarified and revised as

'*Even with similar O/C ratios, the molecular compositions of organics in aerosols and cloud water*
30    *could be quite different, for example, the organosulfate hydrolysis and nitrogen-containing compounds formation were observed in cloud water compared to atmospheric particles, suggesting the significant role of cloud processing in changing the chemical properties of aerosols* (Boone et al., 2015).'

p. 2, l. 31: There are many observational studies that have shown sulfate formation in clouds in the 1980 and 1990s. At least some should be referenced here.

35    **Response**: Yes. The sulfate formation in clouds and its contribution to droplet-mode aerosols have been intensively studied since the 1980s. We have mentioned some of these studies in the first part of the introduction section. Given the emphasis of cloud processing effects on aqSOA in the context, we added

a classic study of in-cloud sulfate formation by Lelieveld and Heintzenberg (1992) and more recent study by Harris et al. (2013) in the beginning and here, and modified text reads,

'In-cloud sulfate production, which causes acid rain, has been extensively characterized *(Lelieveld and Heintzenberg, 1992*; Harris et al., 2013; Guo et al., 2012).'

*'In addition to the in-cloud sulfate formation (Meng and Seinfeld, 1994), the in-cloud organics formation is also likely to add substantial mass to droplet-mode particles (Ervens et al., 2011).'*

Lelieveld, J.; Heintzenberg, J., Sulfate Cooling Effect on Climate through in-Cloud Oxidation of Anthropogenic SO2. Science 1992, 258 (5079), 117-120.

p. 4, l. 3: Define AIM. Did you use the AIM model for any calculation? If so, for what?

**Response**: AIM was an abbreviation of ambient ion monitor, instead of the thermodynamic Aerosol Inorganics Model. To avoid misunderstanding, the abbreviation AIM now is removed and full names are used in the text.

p. 5, l. 28: Here and throughout the manuscript: Numbers should be rounded to their significant digits. For example, here: 96±3, instead of 96.4±2.6.

**Response**: The numbers were rounded to their significant digits as suggested.

p. 7, l. 22: Does this sentence refer to your analysis or to the one by Harris?

**Response**: It refers to our analysis results. This sentence was revised as,

*'Transition metals of Fe, Cu and Mn, which play important roles in the heterogeneous catalytical formation of sulfate (Harris et al., 2013), were also found to be abundant in this study with mean concentrations of 50.6, 10.0 and 5.9 μg L$^{-1}$, respectively.'*

p. 10, l. 7-10: The discussion of the F/A ratios is redundant here and should be included either here or on p. 7 only.

**Response**: All discussion of the F/A ratios was moved to page 10, as mentioned in the above response to comment #4a.

p. 10, l. 17: Are the inverse-power fits empirical or do they have any physical meaning?

**Response**: The inverse-power fits are empirical, which describe the effects of LWC and pH on solute concentrations in cloud water. It has been clarified in the revised text.

p. 11, l. 8 and 10: Add 'carbonyl' here to specify the species group.

**Response**: Added.

p. 14, l. 15: Add a reference for non-radical oxidation of glyoxal.

**Response**: As discussed by Lim et al. (2010) and references therein, aqueous-phase glyoxal can be oxidized by both radical and non-radical reactions. The latter includes hemiacetal formation, aldol condensation, oligomers formation, etc. Here we also cited the laboratory work by Gomez et al. (2015) which revealed the heterogeneous reaction mechanism of glyoxal involving hydration followed by

oligomer formation. We removed the reference (Sui et al., 2017) and adjusted the positions of references as follows,

'Many laboratory experiments have demonstrated that radical (mainly ·OH) *(Lee et al., 2011; Schaefer et al., 2015)* and non-radical aqueous oxidation of glyoxal *(Lim et al., 2010; Gomez et al., 2015)* can produce abundant small carboxylic acids (e.g., oxalic and formic acids), oligomers and highly oxidized organics, which subsequently lead to mass increase in SOA upon droplet evaporation (Galloway et al., 2014).'

p. 15, l. 20: Direct photolysis processes are likely less efficient. 'Photolysis reactions' should be replaced by 'photochemical reactions'.

**Response**: Replaced.

p. 16, l. 5: Oligomer formation is most likely not important in clouds and thus not relevant here, e.g. (Lim et al., 2010)

Response: Thanks for the suggestion. The text is revised as,

*'Although aqueous-phase oligomers can be formed at nighttime, the oligomer formation is most likely not important in clouds (Lim et al., 2010) and thus not relevant here.'*

p. 16, l. 1: During E.4, the sum of sulfate and DOM formation rates (0.52 + 0.45 ug /m3/h) almost equal the total growth rate (1.07 ug/m3/h). However, during E.5 the discrepancy is much greater (0.05+ 0.03 ug/m3/h vs 0.35 ug/m3/h). Can you comment on possible reasons?

**Response**: The rough calculation of net growth rates of DOM and sulfate in cloud water as well as ambient $PM_{2.5}$ was intended to show the trend of aerosol mass growth related to cloudwater sulfate and organics production qualitatively but not quantitatively. Many chemical and physical processes such as cloud scavenging, aqueous oxidation, wet deposition and external air mass intrusion can impact the net change of cloudwater compositions and aerosols. In addition, not all the produced cloudwater sulfate and organics remain in the particulate phase after cloud cycling and contribute to aerosol mass. So the sum of cloudwater sulfate and DOM formation rates is not necessarily equal to the $PM_{2.5}$ growth rate. We expect that the much lower sulfate and DOM formation rates than $PM_{2.5}$ could be ascribed to the intrusion of air masses with elevated $SO_2$ and aerosol particles at the end of cloud event, as well as the artificial calculation of the $PM_{2.5}$ growth rate due to its coarse time resolution of one hour.

p. 16, l. 14: Is this sentence a contradiction of the previous one where you discuss a decrease in DOM?

**Response**: As shown in Figure 7, the nighttime decrease in cloud water DOM corresponds to the decreased glyoxal and carboxylic acids in case E.5, which is consistent with the less decrease in sulfate. However, the DOM/sulfate ratios only slightly decreased from 1.19 to 0.92. To represent the similarly lower DOM/sulfate ratios at lower pH for case E.4 and E.5, we stated the DOM/sulfate ratios to be nearly constant at ~1.0, which is not contradictory to the decrease in DOM as well as sulfate.

Technical comments

p. 2, l. 18: 'contained' can be removed

**Response**: Removed.

p. 2, l. 19: …was three times…

**Response**: Corrected.

p. 4, l. 19: Text is unclear 'into 2 ml a high-pressure liquid chromatography (HPLC) grade acetonitrile'.

**Response**: It was revised to be '*transferred into a volumetric flask using 2 ml high-pressure liquid chromatography (HPLC) grade acetonitrile.*'

p. 5, l. 12: average LWC

**Response**: The 'averaged' was corrected to be 'average' here and throughout the manuscript.

p. 5, l. 22: at many sites

**Response**: Corrected.

p. 7, l. 29: Three-day back trajectories

**Response**: Corrected.

Figure 1: The red cross is very hard to see. Replace by a different color with more contracts (e.g. black) and/or increase the symbol size.

**Response**: The red cross was replaced by a black one with a larger size.

p. 10, l. 19/20: These two sentences need some grammatical corrections.

**Response**: These sentences were revised as '*Unexpectedly, individual carbonyl compounds showed different relationships with LWC and pH. For instance, as LWC and pH increased, glyoxal concentrations decreased in a power function while methylglyoxal tended to increase linearly*.'

p. 10, l. 22:…each carbonyl compound …

**Response**: Corrected.

p. 14, l. 4: Ranges should be presented as $0.72 < r < 0.94$ etc

**Response**: The presentation of ranges throughout the manuscript was revised as suggested.

p. 15, l. 2: 'unsaturation' should be 'subsaturation'

**Response**: Corrected.

[revised manuscript text omitted]

---

## Author Comment (AC2) · 9 Sep 2019

**Response to Referee #2**

General Comments: This study reports on field measurements of trace gases, aerosols, and cloud water at a mountaintop site in Hong Kong. A valuable set of results are provided that are important for the research community interested in cloud processes, especially aqueous processing in clouds. The paper is written fairly well and the methods used seem sound. The conclusions are supported by the data. I did not find too much to comment on in terms of issues and it is my opinion that the paper was constructed well. I only have minor comments below that should be addressed prior to publication.

**Response:** We thank the reviewer for the helpful suggestions on our manuscript. We have made all of the suggested changes and clarifications. The reviewer's comments are in black and our responses are in blue, and the changes in the manuscript are in *italic*.

Specific Comments: Page 15, Line 23-26: This study showed how the oxalate:sulfate ratio grows in clouds and worth noting here for the discussion:

Wonaschuetz, A., et al. (2012). Aerosol and gas re-distribution by shallow cumulus clouds: an investigation using airborne measurements, J. Geophys. Res., 117, D17202, doi:10.1029/2012JD018089.

**Response:** Thanks for the suggestions. We have read this article carefully and cited it in the revision. In addition, the oxalic/sulfate ratio was checked and corrected, which increased from 0.04 to 0.09 as cloud processed and solar radiation intensified. The revised sentence is as follows.

'*The oxalate/sulfate ratio can be indicative of the in-cloud oxalate formation relative to sulfate. For example, aircraft observations (Sorooshian et al., 2007; Wonaschuetz et al., 2012) have shown an increasing aerosol oxalate/sulfate ratio throughout the mixed cloud layer from 0.01 for below-cloud aerosols to 0.09 for above-cloud aerosols, suggesting more aqueous production of aerosol oxalate relative to sulfate by chemical cloud processing.*'

Page 18, Line 19-21: This proposal is also supported by the following and can be added in the discussion to support the authors' speculation: Ervens, B., et al. (2018), Is there an aerosol signature of chemical cloud processing? Atmos. Chem. Phys., 18, 16099-16119, doi: 10.5194/acp-18-16099-2018.

**Response:** Thanks for the suggestions. We read through this paper and learned that the increased mass in larger (droplet-mode) particles could be a signature of chemical cloud processing due to in-cloud sulfate and aqSOA formation. So we cited this paper in the discussion and the texts as follows,

"*Model simulations have revealed that the relative mass increase of droplet-mode aerosols after cloud processing could be up to ~100% for marine air masses with significantly accumulated sulfate and oxalate at 0.56 μm range (Ervens et al., 2018). Hence we can expect that sulfate (air equivalent concentration of 4.9 μg m$^{-3}$) and the low-volatile fraction of DOM (15.0 μg m$^{-3}$) measured in cloud water are mostly retained in droplet-mode aerosols upon cloud evaporation, contributing to the droplet-mode mass fraction. Although the mass size distributions of particle compositions were not measured in this study, the abundant droplet-mode oxalate, organic carbon and sulfate aerosols reported in Hong Kong (Bian et al., 2014; Gao et al., 2016) seem to support our hypothesis.*"

Technical Comments: Page 11: "organics" is spelled wrong

**Response:** The typo is corrected.

---

## Referee Report (RR1)

The authors have addressed most of my previous comments. There are still a couple major issues that needs to be clarified. The first one left me more confused than before. In addition, some minor comments should be addressed, too.

Major comments

1) The authors state in their response and also ion the revised manuscript (p. 4, l. 5) "During the cloud event, the collected fine aerosols were most 5 of the interstitial aerosols, together with some residual particles of smaller droplets < 3 μm."

By definition, interstitial particles are those particles that are not activated into cloud droplets, cf, for example in standard textbooks, e.g. (Seinfeld and Pandis, 2006)

*p. 790: ... The maximum supersaturation reached inside a cloud/fog is an important parameter. Particles with critical supersaturations lower than this value will become activated and become cloud droplets. The rest remain close to equilibrium but never grow enough to be considered droplets and are called interstitial aerosol.*

*p. 794: First, a fraction of the aerosol distribution is activated and becomes cloud droplets while the rest remains as interstitial particles.*

Thus, interstitial aerosol particles are usually too small and/or of too low hygroscopicity to act as nuclei for droplets. Thus, they do not become modified by chemical processes in cloud droplets. Interstitial aerosol usually does not make up a major mass fraction but a major particle fraction. However, only collecting drop residues from drops with a size of 3 um will miss a major fraction of the droplets, i.e. those particles that are likely most significantly altered by chemical cloud processing. It has to be clarified what exactly you collected and how it was done.
As it is written now, i.e., discussing significant sulfate and DOM formation in interstitial aerosol resulting in droplet mode particles is not consistent with the concept of chemical cloud processing which occurs in droplets.

2) The authors claim that the subsaturation of the small monoaldehydes is likely a consequence of their fast oxidation in the aqueous phase. However, comparing the rate constants with OH in the aqueous phase for the compounds listed in Table S4 (all data taken from CAPRAM (Herrmann et al., 2005), I do not see any significant difference that could explain the trends in partitioning.

|  | $k_{OH(aq)}$ / $M^{-1}$ $s^{-1}$ |
|---|---|
| Glyoxal | 1.1e9 |
| Methylglyoxal | 7.9e8 |
| Formaldehyde | 1e9 |
| Acetaldehyde | 3.6e9 |
| Acetone | 1.7e8 |
| Propanal | 2.2e9 |

Thus, I suggest removing the rather vague statement regarding oxidation reactions as a reason for subsaturation. This would be only true if for a given compound the transport from the gas phase is slower than the consumption reaction in the aqueous phase. However, this effect is more important for highly soluble compounds rather than for less soluble ones (Ervens et al., 2014) so that this cannot explain either the observed trend in partitioning.

Minor comments

p., 1, l. 14 (and later in the manuscript): It should be 'acetone' – unless you also measured its substituted forms.

p. 7, l. 16: This is much closer to a factor of 6 not 5 (0.08 vs 0.47).

p. 7, l. 25: 'tens' should be 'ten'
p. 10, l. 26: 'than 1.0' should be removed here

p. 12, l. 10: The study by Waxman et al was also performed for ambient particles. I suggest rewording this sentence to

*'...although the salting-in/out effect is of particular importance in the effective uptake of carbonyl compounds by ambient particles (Shen et al., 2018; Waxman et al., 2013) where solute concentrations are high.'*

p. 13, l. 3: *"In contrast, the less-soluble monocarbonyls are still supersaturated."* This sentence seems redundant as the same is discussed on the previous page.

p. 15, l. 3'''performed'should be replaced by 'shows' or 'exhibits'

p. 15, l. 17: *"Cloud processing can efficiently remove aerosol particles from the air by nucleation scavenging and impaction scavenging, especially at the initial stage of cloud events."* This text is not correct. The scavenged particles are still in the air and not all scavenging processes will result in precipitation.

p. 15, l. 18/19: *"At the same time, chemical cloud processing greatly favors the in-cloud formation of sulfate."* This sentence is not clear either. 'At the same time' implies that you are still talking about the initial stages of cloud events; however, it is more likely that sulfate formation occurs more efficiently in grown droplets, i.e. in later stages. Do you simply want to say that sulfate formation in clouds is globally more significant than in the gas phase?

p. 17, l. 4/5: While you explained in the response that and why the sulfate and DOM loss rates are not additive, this should be also added here as otherwise it is not clear why the sum of the individual loss rates (0.47 ug/m3h + 0.28 ug/m3/h) exceeds the total PM loss rate (0.46 ug/m3h).

p. 17, l. 19: Usually sulfate formation in clouds occurs under oxidant (i.e. $H_2O_2$)-limited conditions, e.g. (Ervens, 2015). Even in the cited study, Shen et al. (2012) state that "The measured aqueous cloud water $H_2O_2$ concentration was not considered appropriate to use in these calculations because it can be rapidly consumed by reaction with S(IV)…"
A better reasoning of the lack of a significant competition for H2O2 by carbonyls vs SO2 is the study by Schöne and Herrmann (2014) that showed that the reactions of carbonyls with H2O2 are likely negligible under cloud conditions.

p. 21, l. 10: The contributions by H. Herrmann should be added to the list.

References

Ervens, B.: Modeling the Processing of Aerosol and Trace Gases in Clouds and Fogs, Chem. Rev., 115(10), 4157–4198, doi:10.1021/cr5005887, 2015.

Ervens, B., Sorooshian, A., Lim, Y. B. and Turpin, B. J.: Key parameters controlling OH-initiated formation of secondary organic aerosol in the aqueous phase (aqSOA), J. Geophys. Res. - Atmos., 119(7), 3997–4016, doi:10.1002/2013JD021021, 2014.

Herrmann, H., A. Tilgner, P. Barzaghi, Z. Majdik, S. Gligorovski, L. Poulain and A. Monod: Towards a more detailed description of tropospheric aqueous phase organic chemistry: CAPRAM 3.0, Atmos. Environ, 39, 4351–4363, 2005.

Schöne, L. and Herrmann, H.: Kinetic measurements of the reactivity of hydrogen peroxide and ozone towards small atmospherically relevant aldehydes, ketones and organic acids in aqueous solutions, Atmos. Chem. Phys., 14(9), 4503–4514, doi:10.5194/acp-14-4503-2014, 2014.

Seinfeld, J. H. and Pandis, S. N.: Atmospheric Chemistry and Physics - From air pollution to climate change, 2nd ed., edited by John Wiley, John Wiley & Sons, Inc., Hoboken, New Jersey., 2006.

Shen, H., Chen, Z., Li, H., Qian, X., Qin, X. and Shi, W.: Gas-Particle Partitioning of Carbonyl Compounds in the Ambient Atmosphere, Environ. Sci. Technol., 52(19), 10997–11006, doi:10.1021/acs.est.8b01882, 2018.

Shen, X., Lee, T., Guo, J., Wang, X., Li, P., Xu, P., Wang, Y., Ren, Y., Wang, W., Wang, T., Li, Y., Carn, S. A. and Collett Jr, J. L.: Aqueous phase sulfate production in clouds in eastern China, Atmos. Environ., 62(0), 502–511, doi:http://dx.doi.org/10.1016/j.atmosenv.2012.07.079, 2012.

Waxman, E. M., Dzepina, K., Ervens, B., Lee-Taylor, J., Aumont, B., Jimenez, J. L., Madronich, S. and Volkamer, R.: Secondary organic aerosol formation from semi- and intermediate-volatility organic compounds and glyoxal: Relevance of O/C as a tracer for aqueous multiphase chemistry, Geophys. Res. Lett., 40, 1–5, doi:10.1002/grl.50203, 2013.

---

## Author Response (AR2)

The authors have addressed most of my previous comments. There are still a couple major issues that needs to be clarified. The first one left me more confused than before. In addition, some minor comments should be addressed, too.

**Response:** We thank the reviewer for the helpful comments. We have made all of the suggested changes and clarifications. The reviewer's comments are in black and our responses are in blue, and the changes in the manuscript are in blue *italic*.

Major comments

1) The authors state in their response and also ion the revised manuscript (p. 4, l. 5) "During the cloud event, the collected fine aerosols were most of the interstitial aerosols, together with some residual particles of smaller droplets < 3 μm."

By definition, interstitial particles are those particles that are not activated into cloud droplets, cf, for example in standard textbooks, e.g. (Seinfeld and Pandis, 2006)

*p. 790: ... The maximum supersaturation reached inside a cloud/fog is an important parameter. Particles with critical supersaturations lower than this value will become activated and become cloud droplets. The rest remain close to equilibrium but never grow enough to be considered droplets and are called interstitial aerosol.*

*p. 794: First, a fraction of the aerosol distribution is activated and becomes cloud droplets while the rest remains as interstitial particles.*

Thus, interstitial aerosol particles are usually too small and/or of too low hygroscopicity to act as nuclei for droplets. Thus, they do not become modified by chemical processes in cloud droplets. Interstitial aerosol usually does not make up a major mass fraction but a major particle fraction. However, only collecting drop residues from drops with a size of 3 um will miss a major fraction of the droplets, i.e. those particles that are likely most significantly altered by chemical cloud processing. It has to be clarified what exactly you collected and how it was done.

As it is written now, i.e., discussing significant sulfate and DOM formation in interstitial aerosol resulting in droplet mode particles is not consistent with the concept of chemical cloud processing which occurs in droplets.

**Response**: We thank the reviewer for the valuable suggestions. We agree that the interstitial aerosol particles are usually too small or of too low hygroscopicity to act as nuclei for droplets, in that case, the interstitial particles usually do not become modified by chemical processes in cloud droplets, and it may not be appropriate to discuss the cloud processing effect on the interstitial aerosols.

First, we want to say that the interstitial aerosol particles still could account for a large fraction both in mass and particle number, in the polluted environment with much higher particulate matter concentrations, such as in China. For example, high $PM_{2.5}$ concentrations (~5−50 μg m$^{-3}$) remain as interstitial particles during cloud periods observed at Mt. Tai in north China (Li et al., 2017). Single-particle analysis at Mt. Tai (Figure R1) indicated that the median diameters of cloud residual

and interstitial particles were 1.19 µm and 422 nm, respectively (Liu et al., 2018). Interstitial particles with larger sizes are likely existing in the polluted cloud events, compared to most of the previous studies in relatively clean environment. Although they were not activated and did not grow to cloud droplets, there were still thin water layers absorbed by the particle surface (Figure R1), and thus aqueous reaction may still play important roles in affecting the chemical properties of the wetted interstitial particles. A field observation at a remote mountain site in southern China using single-particle mass spectrometry (SP-MS) suggested that in-cloud aqueous reactions could enhance oxalate particle formation in both the cloud residual and interstitial particles, in comparison with the nearly constant oxalate in cloud-free particles (Figure R2) (Zhang et al., 2017).

[Figure]

**Figure R1.** TEM images showing the mixing properties of multicomponent within individual cloud residual (top panel) and interstitial (bottom panel) particles (Liu et al., 2018).

[Figure]

**Figure R2.** Size-dependent number fractions of oxalate-containing particles relative to all the detected cloud-free, cloud residual and cloud interstitial particles (Zhang et al., 2017).

In the present study, we could not make a conclusion and direct comparison of the aqueous reactions between the interstitial particle and cloud droplets due to the lack of high time-resolution and direct measurement. The importance of aqueous processing on interstitial particles needs further comprehensive investigations in future works.

In the methodology section, we have clarified the text as,

'*During the cloud event, the collected fine aerosols were most of interstitial aerosols, because almost all activated cloud droplets were larger than 3 μm and were removed by the cyclone in the sampling inlet.*'

In the results section, we remove the discussion on in-cloud interstitial particles in the revised manuscript, to avoid causing misunderstanding. Instead, we compare the post-cloud aerosols with the pre-cloud aerosols to examine the cloud processing effects on aerosol properties after droplets evaporation. Due to the limited valid data of ambient inorganic ions and particle mass size distributions, we only compared the chemical compositions of pre- and post-cloud aerosols in E.1 and E.3 in Section 3.4.2.

Figure 9 and S6 has also been revised as shown below, and the related texts were revised as follows:

[revised manuscript text omitted]

2) The authors claim that the subsaturation of the small monoaldehydes is likely a consequence of their fast oxidation in the aqueous phase. However, comparing the rate constants with OH in the aqueous phase for the compounds listed in Table S4 (all data taken from CAPRAM (Herrmann et al., 2005), I do not see any significant difference that could explain the trends in partitioning.

|  | $k_{OH(aq)}$ / $M^{-1}s^{-1}$ |
| --- | --- |
| Glyoxal | 1.1e9 |
| Methylglyoxal | 7.9e8 |
| Formaldehyde | 1e9 |
| Acetaldehyde | 3.6e9 |
| Acetone | 1.7e8 |
| Propanal | 2.2e9 |

Thus, I suggest removing the rather vague statement regarding oxidation reactions as a reason for subsaturation. This would be only true if for a given compound the transport from the gas phase is slower than the consumption reaction in the aqueous phase. However, this effect is more important for highly soluble compounds rather than for less soluble ones (Ervens et al., 2014) so that this cannot explain either the observed trend in partitioning.

**Response**: Thanks for pointing out this and the helpful suggestions. Based on the listed comparable rate constants for individual carbonyls with OH in the aqueous phase, oxidation reactions are likely unable to explain the trends in partitioning neither supersaturation nor subsaturation. The related revisions were made as follows,

'Formaldehyde was deficient in the cloud water with a $F_{me}/F_{theo}$ value of 0.12, *similar to the lower measured formaldehyde in the aqueous phase than that expected at equilibrium reported by (Li et al., 2008)*, who suggested that it was probably associated with aqueous oxidation of formaldehyde.'

'The complicated partitioning behaviors could be affected by both physical (e.g., interface adsorption effect) and chemical processes (e.g.,  *chemical production from precursors*) (van Pinxteren et al., 2005, and references therein).'

''

'The complicated partitioning behaviors of carbonyls possibly result from the *combined physical and chemical effects*, which require further investigation.'

Minor comments

p., 1, l. 14 (and later in the manuscript): It should be 'acetone' – unless you also measured its substituted forms.

**Response**: Only acetone was measured here. It was corrected to be 'acetone'.

p. 7, l. 16: This is much closer to a factor of 6 not 5 (0.08 vs 0.47).

**Response**: Changed.

p. 7, l. 25: 'tens' should be 'ten'

**Response**: Corrected.

p. 10, l. 26: 'than 1.0' should be removed here

**Response**: Removed.

p. 12, l. 10: The study by Waxman et al was also performed for ambient particles. I suggest rewording this sentence to

*'...although the salting-in/out effect is of particular importance in the effective uptake of carbonyl compounds by ambient particles (Shen et al., 2018; Waxman et al., 2013) where solute concentrations are high.'*

**Response**: The suggested sentence is more concise and we have revised it accordingly.

p. 13, l. 3: *"In contrast, the less-soluble monocarbonyls are still supersaturated."* This sentence seems redundant as the same is discussed on the previous page.

**Response**: This redundant sentence was removed.

p. 15, l. 3: 'performed' should be replaced by 'shows' or 'exhibits'

**Response**: The 'performed' was replaced by 'exhibits'.

p. 15, l. 17: *"Cloud processing can efficiently remove aerosol particles from the air by nucleation scavenging and impaction scavenging, especially at the initial stage of cloud events."* This text is

not correct. The scavenged particles are still in the air and not all scavenging processes will result in precipitation.

**Response**: Agree. We want to say that cloud processing can scavenge aerosol particles from the gas phase into droplets, which is one effect of cloud on aerosol particles. The sentence was reworded as

'*Cloud processing can efficiently alter the aerosol concentration and composition* by nucleation and impaction scavenging (Ervens, 2015), especially at the initial stage of cloud events (Wang et al., 2011; Li et al., 2017b).'

p. 15, l. 18/19: *"At the same time, chemical cloud processing greatly favors the in-cloud formation of sulfate."* This sentence is not clear either. 'At the same time' implies that you are still talking about the initial stages of cloud events; however, it is more likely that sulfate formation occurs more efficiently in grown droplets, i.e. in later stages. Do you simply want to say that sulfate formation in clouds is globally more significant than in the gas phase?

**Response**: In this sentence, we want to express another important cloud effect on aerosol particles apart from the scavenging effect, i.e. the in-cloud formation of sulfate. As suggested, the use of 'at the same time' is not appropriate. So we reworded the sentence as

'*On the other hand*, chemical cloud processing greatly *favors* the in-cloud formation of sulfate (Harris et al., 2013) and SOA (Brégonzio-Rozier et al., 2016), *which can remain in the particle phase upon droplet evaporation*.'

p. 17, l. 4/5: While you explained in the response that and why the sulfate and DOM loss rates are not additive, this should be also added here as otherwise it is not clear why the sum of the individual loss rates (0.47 ug/m3h + 0.28 ug/m3/h) exceeds the total PM loss rate (0.46 ug/m3h).

**Response**: According to the major comment 1# above, we think that the $PM_{2.5}$ measured during cloud events were all interstitial aerosols, and it could cause controversy or misunderstanding to compare the interstitial aerosol changes with the cloud water constitutes. Further, the losses of cloud water sulfate and DOM were not proved to directly induce the ambient PM loss in this study. So to make it clear, we revised Figure 7 and removed all the discussion relating to the growth rate of $PM_{2.5}$. The texts were reworded as

'Therefore, photochemical reactions are expected to enhance the cloud ]water organics production (0.51 $\mu g\ m^{-3}\ h^{-1}$) compared to sulfate (0.10 $\mu g\ m^{-3}\ h^{-1}$) in case E.2. During the mixed E.4, the approximate growth rates of DOM (0.52 $\mu g\ m^{-3}\ h^{-1}$) and sulfate (0.45 $\mu g\ m^{-3}\ h^{-1}$) were comparably fast. *By contrast, the growth of daytime DOM (0.05 $\mu g\ m^{-3}\ h^{-1}$) and sulfate (0.03 $\mu g\ m^{-3}\ h^{-1}$) during the marine E.5 was observed much slow, which could be resulted from the relatively fewer precursors in the cleaner marine air masses*. It was noted that aqueous glyoxal gradually increased after sunrise *(6–37 ng $m^{-3}\ h^{-1}$)*, which likely produced carboxylic acids such as oxalic acid rapidly via photo-oxidation and contributed to the formation of aqueous organics (Carlton et al., 2007; Warneck, 2003). *But at nighttime, the decrease of aqueous glyoxal (-6.8 ng $m^{-3}\ h^{-1}$) was observed concurrent with the apparent reduction of DOM (-0.47 $\mu g\ m^{-3}\ h^{-1}$) in E.5*. Although aqueous-phase

oligomers can be formed at nighttime, the oligomer formation is most likely not important in clouds (Lim et al., 2010) and thus not relevant here.'

p. 17, l. 19: Usually sulfate formation in clouds occurs under oxidant (i.e. H2O2)-limited conditions, e.g. (Ervens, 2015). Even in the cited study, Shen et al. (2012) state that "The measured aqueous cloud water H2O2 concentration was not considered appropriate to use in these calculations because it can be rapidly consumed by reaction with S(IV)…"

A better reasoning of the lack of a significant competition for H2O2 by carbonyls vs SO2 is the study by Schöne and Herrmann (2014) that showed that the reactions of carbonyls with H2O2 are likely negligible under cloud conditions.

**Response**: Thanks for the clarification and suggestion. We checked the kinetic study by Schöne and Herrmann (2014) and replaced the reference in the revised text, as follows,

'Additionally, *there is a lack of significant* competition for $H_2O_2$ *by* carbonyl compounds *versus* S(IV) *since the reactions of carbonyls with* $H_2O_2$ *are likely negligible under cloud conditions due to their very small reactivities (Schöne and Herrmann, 2014)*.'

p. 21, l. 10: The contributions by H. Herrmann should be added to the list.

**Response**: H. Herrmann has helped us on the data analysis and provided many comments and suggestions on the data analysis and results interpretation, especially on the carbonyls partitioning and aqueous organics formation. We have added H. Herrmann's contributions in the revised text.

[revised manuscript text omitted]

---

## Author Response (AR3)

**Response to Editor's comments**

Thank you for your revisions. The manuscript is almost ready for publication in ACP after a few minor details have been clarified and/or changed.

**Response:** We thank the editor for the helpful comments. We have made all of the suggested changes and clarifications. The editor's comments are in black and our responses are in blue, and the changes in the manuscript are in blue *italic*.

Page 2, line 28: Please change "contribute to aerosol mass production but also alter the" to "contributes to aerosol mass production but also alters the"

**Response:** Changed.

Page 7, line 7: Please change "was ten of times" to "was around ten times"

**Response:** Changed.

Page 8, line 5: Please add which meteorological data you used for the HYSPLIT model.

**Response:** We added meteorological data information in the caption of Figure 1, as follows,

"Figure 1. Air mass plumes arriving at Mt. TMS (black cross) in Hong Kong for six cloud events (E.1–6) simulated using the HYbrid Single-Particle Lagrangian Integrated Trajectory (HYSPLIT) model. *The model was driven by archive meteorological data from the Global Data Assimilation System (GDAS, 1° × 1° resolution).*"

In Figure 3 (last panel) you show that you receive a linear relationship if you exclude certain datapoints. However, I see no discussion of this in the text. Is this justified? Is it to be expected?

**Response:** Thanks for pointing this out. The relationship between methylglyoxal and LWC is an example to show that the LWC and pH could have different effects on specific carbonyl compounds except for the common dilution and dissolution effects. We found high methylglyoxal concentrations at high LWC and pH conditions and a positive relationship when the three low concentration data points outside of the blue dashed circle were excluded. Different carbonyl compounds also show different relationships (increase or decrease) with LWC and RH, as depicted in Figure 4S, which may be related to its gas/aqueous partitioning behavior. More discussions were added in the revised text, as follows,

'*Similarly, glyoxal concentrations decreased in a power function as LWC and pH increased. However, methylglyoxal* showed different relationships with LWC and pH. *Unexpectedly*, methylglyoxal tended to increase linearly *with LWC and RH for most of the samples except three low-concentration samples (Figure 3d and Figure S4b), which show the opposite pattern to the dilution effect and suggest more methylglyoxal dissolved in diluted and less acidic cloud waters. The different relationship (increase or decrease) of* other carbonyls concentrations with increased LWC and pH are also shown in Figure S4.'

Figure 3 caption was reworded as '…Methylglyoxal has *a good linear relationship with LWC for samples in the* blue dashed *circle*.'

In Figure 6, you are presenting ellipses which are supposed to represent the "confidence coefficient of 99%". Could you please give a reference on what this actually parameter means.

**Response:** Thanks for pointing out this typo, it should be the "99% confidence level".

The confidence ellipse is a typical way to visualize the population means of bivariate normal distribution data and their variation, examining the correlation between two variables. As confidence curves, the ellipses show where the specified percentage of the data should lie, assuming a bivariate normal distribution. As visual indicators of correlations, the confidence ellipse collapses diagonally as the correlation between two variables approaches 1 or -1. The confidence ellipse is more circular when two variables are uncorrelated. In the present study, the 99% confidence ellipses were plotted using the OriginPro 2015 software.

Figure 6 caption was modified to make it clearer.

'Figure 6. Pairwise scatter plot of selected organic species in cloud water. The confidence ellipses *were plotted (using OriginPro 2015 software) to indicate the strength of bivariate correlations at 99% confidence level. The confidence ellipse collapses diagonally as the correlation between two variables approaches 1 or -1, and become more circular when two variables are uncorrelated.*'

Page 19, line 1: Please remove the "was" after "fraction".

**Response:** Removed.

Figure 10: What particle density did you use to illustrate the mass size distributions? I could not find this number in the text.

**Response:** In the present study, the particle density of 1.65 $\mu g\ m^{-3}$ was used to calculate the mass size distributions. We added this number to the caption of Figure 10.

Concerning the data availability: Copernicus requests that all data should be made available on reliable (public) data repositories to assure open and reproducible science. There are numerous accessible databases available. If the data are not publicly accessible, a detailed explanation of why this is the case is required. Please consult https://www.atmospheric-chemistry-and-physics.net/about/data_policy.html

**Response:** All the field measurement data are available by contacting the corresponding author.

[revised manuscript text omitted]